# SFC active reconfiguration based on user mobility and resource demand prediction in dynamic IoT-MEC networks

Shuang Guo[1,2], Liang Liu [ID][3]*, Tengxiang Jing[3], Huan Liu[4]*

**1** Chongqing College of Mobile Communication, Qijiang, Chongqing, China, **2** Chongqing Key Laboratory of Public Big Data Security Technology, Qijiang, Chongqing, China, **3** School of Communication and Information Engineering, Chongqing University of Posts and Telecommunications, Chongqing, China, **4** Department of Orthopedics, The Second Affiliated Hospital of Army Medical University, Chongqing, China

\* liuliang@cqupt.edu.cn (LL); xq_liuhuan@tmmu.com.cn (HL)

**Data Availability Statement:** The data set of trajectory prediction model presented in the study are available from http://research.microsoft.com/en-us/downloads/b16d359d-d164-469e-9fd4-daa38f2b2e13/, http://www.cs.uic.edu/%7Eboxu/

## Abstract

To achieve secure, reliable, and scalable traffic delivery, request streams in mobile Internet of Things (IoT) networks supporting Multi-access Edge Computing (MEC) typically need to pass through a service function chain (SFC) consisting of an ordered series of Virtual Network Functions (VNFs), and then arrive at the target application in the MEC for processing. The high mobility of users and the real-time variability of network traffic in IoT-MEC networks lead to constant changes in the network state, which results in a mismatch between the performance requirements of the currently deployed SFCs and the allocated resources. Meanwhile, there are usually multiple instances of the same VNF in the network, and proactively reconfiguring the deployed SFCs based on the network state changes to ensure high quality of service in the network is a great challenge. In this paper, we study the SFC Reconfiguration Strategy (SFC-RS) based on user mobility and resource demand prediction in IoT MEC networks, aiming to minimize the end-to-end delay and reconfiguration cost of SFCs. First, we model SFC-RS as Integer Linear Programming (ILP). Then, a user trajectory prediction model based on codec movement with attention mechanism and a VNF resource demand prediction model based on the Long Short-Term Memory (LSTM) network are designed to accurately predict user trajectories and node computational and storage resources, respectively. Based on the prediction results, a Prediction-based SFV Active Reconfiguration (PSAR) algorithm is proposed to achieve seamless SFC migration and routing update before the user experience quality degrades, ensuring network consistency and high quality service. Simulation results show that PSAR provides 51.28%, 28.60%, 21.75%, and 16.80% performance improvement over the existing TSRFCM, DDQ, OSA, and DPSM algorithms in terms of end-to-end delay reduction, and 33.32%, 18.94%, 67.42%, and 60.61% performance optimization in terms of reconfiguration cost reduction.

mp2p/gps%5Fdata.html). The data set of load prediction model presented in the study are available from GWA-T-12 Bitbrains: http://gwa.ewi.tudelft.nl/datasets/gwa-t-12-bitbrains/.

**Funding:** This research was funded by the Science and Technology Research Youth Project of Chongqing Municipal Education Commission under Grant number KJQN202202401, Chongqing University of Posts and Telecommunications Doctoral Initiation Foundation under Grant number A2023007, Joint Medical Research Program of Chongqing Municipal Science and Technology Bureau and Chongqing Municipal Health Commission under Grant number 2024QNXM010. The funders had no role in study design, data collection and analysis, decision to publish, or preparation of the manuscript.

**Competing interests:** The authors have declared that no competing interests exist.

# Introduction

The rapid development of Internet of Thing (IoT) technology has resulted in more and more heterogeneous computing intensive and delay sensitive service request flows at IoT terminals such as industrial sensors, smart cameras, etc. [1]. In order to meet the security, reliability, time sensitivity requirements of these request flows, Internet Service Providers (ISPs) can use Multi-access Edge Computing (MEC) technology [2] and Network Function Virtualization (NFV) technology [3] to deploy a Service Function Chain (SFC) composed of multiple ordered Virtual Network Functions (VNFs) at the edge of the network to provide high-quality network services for these request flows [4, 5].

Generally, in IoT-MEC networks, the high mobility of users can potentially result in the switching of user edge access location, thereby inducing previously deployed SFCs for user to violate end-to-end latency constraints [6–8]. Additionally, the continuously increasing number of IoT devices are attempting to access edge services at anytime and anywhere, this will lead to real-time fluctuations of the network traffic [9–11]. The dynamic scenario can cause the resource demands of the deployed SFCs on the underlying network to change, and it may lead to violate the resource usage threshold of the deployment node. In order to guarantee the Quality of Service (QoS) of network and ensure to provide consistent high-quality service for users, ISPs need to reconfigure or seamlessly migrate some deployed SFCs in the network [12, 13]. However, due to the rapid real-time changes of network traffic and the high mobility of users in the IoT-MEC network, how and when to reconfigure SFC is a great challenging problem [14].

There have been studies focusing on SFC deployment strategies aimed at reducing operational costs, improving resource utilization, and guaranteeing quality of service (QoS). Passive reconfiguration studies respond to changes in the network environment, but can lead to frequent migrations and instability. Active reconfiguration preempts migrations before QoS degradation, avoids service disruptions, and improves performance and user experience. Specific studies reduce migration time based on user movement predictions. Liu et al. [15] used Graph Neural Network (GNN) to predict resource demand and applied deep Dyna-Q algorithm to handle SFC reconfiguration in IoT. And [16] predicted resource demand through federated learning-based Bidirectional Gated Recurrent Unit (Bi-GRU) algorithm and combined it with Deep Reinforcement Learning (DRL) for VNF migration decision. These studies reduce the number and cost of VNF migrations, but do not consider the impact of user mobility on migration performance, Especially, there are few studies which jointly consider user mobility and resource demand to deal with SFC reconfiguration.

Different from the above researches, in this paper, we investigate the SFC Reconfiguration Strategy (SFC-RS) in IoT-MEC networks jointly considering user mobility and resource demand prediction aiming at minimizing the end-to-end delay and reconfiguration cost of SFC. First, we model the SFC-RS as an Integer Linear Programming (ILP). We then designed the Encoder-Decoder mobile user trajectory prediction model based on attention mechanism and the VNF resource demand prediction model based on Long Short-Term Memory (LSTM) network to predict the user trajectory and node load, respectively. According to the prediction results, we devise a Prediction based SFC Active Reconfiguration (PSAR) algorithms to achieve the seamless migration and routing update of SFC before the the network QoS degrading. Simulation results show that the proposed method can provide for users with consistent high-quality network services. The main contributions of this paper are summarized as follows:

- We define the SFC active reconfiguration problem in IoT-MEC networks and formulate it as an ILP aiming at minimizing the end-to-end delay and reconfiguration cost of IoT SFC services.

- We adopt a data preprocessing method that converts latitude and length to a grid to obtain the user's location while maintaining a certain degree of user privacy and design an Encoder-Decoder trajectory prediction model based on an attention mechanism to predict the user's location.

- We jointly consider that a single VNF instance may be shared by multiple SFCs and design an Long Short-Term Memory (LSTM) based algorithm to predict the load of edge servers in the network.

- According to the above real-time prediction results, we propose a Predict based SFC Active Reconfiguration (PSAR) algorithm to complete the seamless migration of VNFs and the selection of reroute path of SFC.

- We conduct extensive experiments to evaluate our scheme, simulation results show that the PSAR has better performance in the end-to-end delay and reconfiguration cost of the SFC compared with existing algorithms.

The remainder of this paper is organized as follows. Section reviews the related work. The system model and problem definition is presented in Section. In Section, a PSAR algorithm is proposed to tackle the SFC actively seamless migration based on the prediction of user movement trajectory and node load. Furthermore, we analyze the performance of the proposed approaches in Section. This paper is concluded in Section.

## Related work

We classify the related research work of SFC reconfiguration into two categories, they are SFC passive reconfiguration and SFC active reconfiguration.

### SFC passive reconfiguration

Currently, there have been many studies on the SFC deployment strategy [17–23], which focus on how to reduce the operation costs of the network providers, enhance the resource utilization rate of the network, and guarantee network QoS. There are also some studies on the passive reconfiguration of SFC [24–31], which is caused by triggering a certain condition in a variable network environment. Owing to the dynamic changes in resources of each node, user's movement, etc., may cause the rapid change of network state, this will continually trigger the reconfiguration condition and result in frequent migration of SFC, thereby reducing the stability of the network and the quality of user experience. Moreover, the passive SFC reconfiguration mechanism usually has a time lag, in severe cases, this may lead to network service interruption [32, 33].

Specifically, The authors in [24] study the passive reconfiguration scheme of SFC in mobile edge networks to support the seamless migration of services when users move across mobile base stations. They assume that the user's movement trajectory is known in advance, but in practice, the movement trajectory of users is usually randomness, and the method of assuming that the location is known in advance will cause some SFC migration failures. In [25], the authors devise and estimate four SFC passive migration modes for applications depending on time synchronization in 5G Networks and Beyond. The authors in [26] propose a shared file system-based way to reduce the passive migration time of SFC and the service outage time. The above two works mainly take into account the passive migration time of SFC while ignoring the migration cost of SFC. Addad et al. [27] use two deep reinforcement learning-based algorithms to detect the required bandwidth for a given SFC request flow, and assign bandwidth resources for the links according to the detection results during passive migration of

SFCs, their proposed AI agent ignores the resources detection of network nodes, and their work also does not consider the migration cost of SFCs.

In [30], the authors propose a mobile aware SFC migration scheme in the MEC scenario with small base stations. They firstly classify users into two categories, they then conduct SFC migration for users who move to the better Access Point(AP) and may stay there for a longer period of time. For those users who move to a AP with poor service quality, due to the SFC migration may result in frequent AP switching, therefore, these SFCs of users will not be migrated. They finally use a reinforcement learning framework to obtain the optimal SFCs migration strategy from historical experience. Zhang et al. [28] propose an online inert migration adaptive interference detection algorithm for real-time deployment and migration of VNFs in 5G network slices. The authors in [29] introduce a heuristic algorithm and reinforcement learning algorithm to solve the placement and passive migration of SFC in NFV network. These works in [28–30] ignore the impact of real-time dynamic changes of network traffic on server load in the cloudlets, when executing SFC passive migration, these methods may lead to resource fragmentation or waste in the network. [31] investigates the optimal location allocation for concurrent passive migration of VNF instances with the goal of minimizing the end-to-end delay of all affected SFCs by network nodes load and ensure the network load balance after VNF migration. Nevertheless, the authors do not consider the migration cost of VNF and the choice of reroute path for SFC after VNFs migration.

## SFC active reconfiguration

Conversely, the SFC active reconfiguration mechanism can complete the active seamless migration and routing updates of VNF in advance before the network QoS declines, this solves the lag deficiency of passive SFC reconstruction strategy, thereby ensuring the high performance and user quality of experience of the network. Specifically, the authors in [34] investigate the SFC migration timing decision problem based on the known user movement path and predicted arrival time. They establish a predictive model to forecast user arrival time. The experimental results show that their method can reduce the average VNF migration time. Nevertheless, the user's movement trajectory is known in advance is usually unrealistic. The authors in [35] proposes a Topology-aware Min-latency SFC Migration (TMSM) method to strike a desirable balance between the SFC latency and the migration cost. They apply a Markov approximation based heuristic to seek a near-optimal solution for SFC migration. The evaluations show that the TMSM achieves a better tradeoff between the SFC latency and migration cost. However, the authors don't consider actively predicting the resource load of nodes in the network, it results in a low success rate of SFC migration.

Some works study the active reconstruction strategy of SFC. Particularly, [36] proposes a SFC active reconfiguration mechanism based on node computing load and SFC resource demand prediction [37] studies the SFC migration problem in the core cloud considering the migration cost and the balance of physical resource distribution in the network. They model the SFC migration problem as an ILP and propose an active migration method that can effectively decrease the imbalance of physical resource distribution in the network, and thus improve the acceptance ratio of SFC request flows, physical resources utilization, and the long-term profit of network operators. The above mentioned works have certain advantages on the SFC migration compared with passive SFC reconfiguration, but these methods do not consider the impact of the high mobility of users on the network state, they can not achieve the seamless migration of SFCs. Therefore, in this paper, we investigate the SFC active reconfiguration jointly considering user mobility and node resource demand prediction in the network.

## System model and problem definition

### Physical network

We Model the MEC network as an undirected graph $G = (C \cup B, L)$, where $B$ indicates the set of base stations, $C$ represents the set of Cloudlets, $L$ indicates the set of physical links. As shown in the dashed box at the bottom of Fig 1, ISP run these Cloudlets to provide services for users, and the time interval $T$ is divided into many small cycle intervals of the same size, we call them as time slots. At the beginning of each time slot, mobile users in the network will access the MEC network through the base station and apply for service requests from the ISP. Each base station, as a relay point for mobile users in the MEC network, is located in different areas of the network and only plays the role of service access and traffic forwarding, and the Cloudlet and the base station are in the same position.

Generally, the types of resources on the Cloudlet include CPU, memory and storage resources. We assume that the storage resources on each Cloudlet are sufficient, but the CPU resources and memory resources are limited, for each Cloudlet $c \in C$ in the MEC network, we use $C_c$ and $M_c$ to denote the CPU and memory resources capacity, respectively. The range of computing resource utilization rate is denoted by $\breve{\mu}_c \leq \mu_c \leq \widehat{\mu}_c$, the range of the memory resource utilization rate is denoted by $\breve{\mu}_m \leq \mu_m \leq \widehat{\mu}_m$. The bandwidth capacity on the physical link between base station $u$ and base station $v$ is $B_{uv}$. The base station and the Cloudlet has a high-speed link, we ignore the link delay between them. The main notations related to the

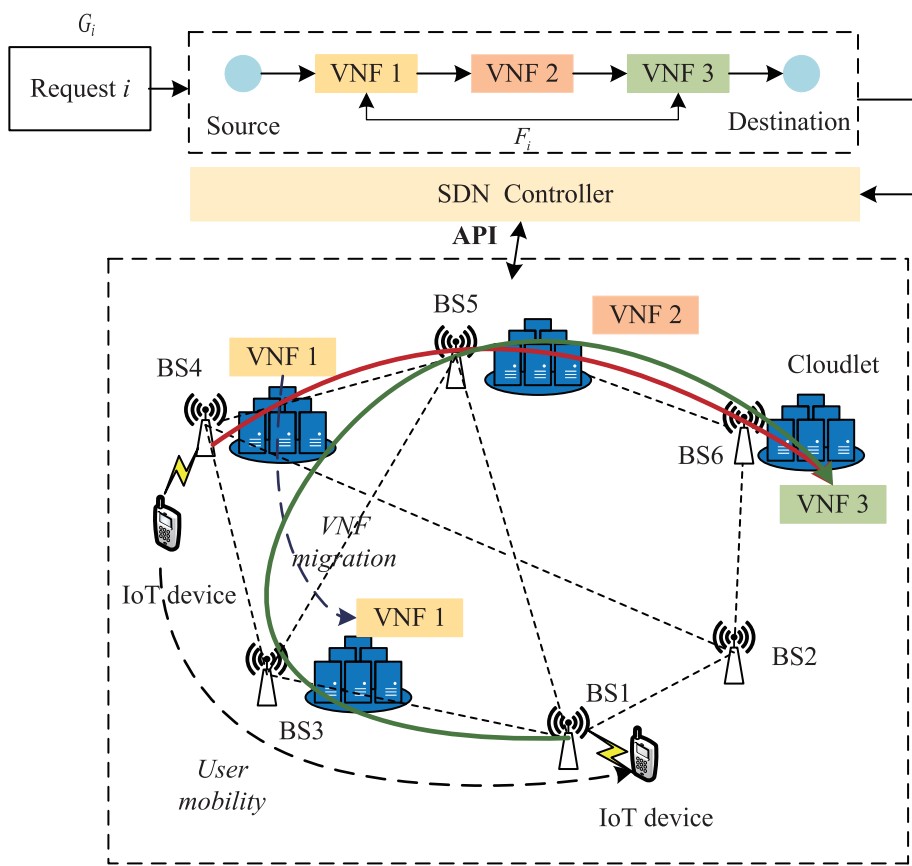

**Fig 1. SFC reconfiguration scenario.**

**Table 1. Notations table.**

| | Physical Network Symbols |
|---|---|
| $G = (C \cup B, L)$ | IoT-MEC network with cloudlets($C$), base stations($B$) and physical links ($L$) |
| $C_c, M_c$ | CPU and memory capacity of the cloudlet ($c \in C$), respectively |
| $\mu_c, \mu_m$ | The resource utilization of CPU and memory, respectively |
| $B_{uv}$ | The bandwidth capacity of the physical link connecting base stations $u$, $v$ |
| $C_c^t, b_l^t$ | The available CPU resources on cloudlet $c$ and the available bandwidth resources on physical link $l$ at time slot $t$, respectively |
| $c_l$ | The propagation cost per 100Mbit of data transmitted on physical link $l$ |
| | **SFC Symbols** |
| $R$ | The set of all IoT request flows in the IoT-MEC network |
| $G_i = (F_i, L_i)$ | The service function graph $G_i$ of IoT request flow $i$ with required VNF set $F_i$ and virtual link set $L_i$ |
| $f_{i,m}^{cpu}, f_{i,m}^{mem}$ | The computing and memory resource requirements of VNF $m \in F_i$ |
| $R_i^{delay}, R_i^{bw}$ | The maximum tolerated delay, bandwidth resource requirements of IoT request flow $i$ |
| $T_i$ | The maximum tolerated delay for SFC seamless migration |
| | **Binary Variables** |
| $x_{i,m}^c$ | denotes whether the IoT request flow $i$ is processed by VNF $m$ on cloudlet $c$ |
| $x_i^{t,c,c'}$ | indicates whether the IoT request flow $i$ will migrate services from cloudlet $c$ to $c'$ at time slot t |
| $y_l^{t,c,c'}$ | indicates whether the data packets migrated by VNF from cloudlet $c$ to $c'$ use physical link $l$ as the reroute path |
| $y_{i,mn}^l$ | indicates whether the virtual link between adjacent VNF $m$ and $n$ of IoT request flow $i$ is mapped to physical link $l$ |

physical network, SFCs, and binary variables are shown in Table 1, and the other variables are shown in Table 2.

## SFC model

We use $R$ to represent the set of all request flows in the IoT-MEC network. As shown in the dashed box at the upper of Fig 1, the $i^{th}$ request flow from a mobile user is modeled as SFC and

**Table 2. Notations table.**

| | Other Variables |
|---|---|
| $d_{que}(i, t), d_{mig}(i, t), d_{trans}(i, t)$ | The queue delay, migration delay and transmission delay of IoT request flow $i$ at time slot $t$, respectively |
| $d_{pro}(i, t)$ | The processing delay of cloudlet $c$ at time slot $t$ |
| $D(i, t), C(i, t)$ | The E2E delay and Reconfiguration cost of IoT request flow $i$ at time slot $t$ |
| $\lambda_m^t, \mu_m^t$ | The data rate of packets of IoT request flow $i$ that arrive at VNF $m$, and service rate of cloudlet $c \in C$ at time slot $t$, respectively |
| $Q_{i,m}^{t,c,c'}$ | The size of the data packets that need to be migrated of IoT request flow $i$ at time slot $t$ |
| $b_{uv}^{t,c,c'}$ | The bandwidth resources of the physical link $uv$ used to migrate services from cloudlet $c$ to $c'$ at time slot $t$ |
| $f_{c',m}^t$ | The processing power of VNF $m$ on cloudlet $c'$ at time slot $t$ |
| $Q_{i,m}^t$ | The packet arrival amount on VNF $m$ at time slot $t$ |
| $b_{i,mn}^{t,uv}$ | The bandwidth resources on the physical link mapped to by the virtual link $mn$ between adjacent VNFs of IoT request flow $i$ at time slot $t$ |
| $v_{i,m}^t$ | The cloudlet hosted by VNF instance $m$ of IoT request flow $i$ at time slot $t$ |
| $P_{i,t}^m$ | The migration paths set of VNF instance $m$ of IoT request flow $i$ at time slot $t$ |
| $Q_{i,m,l}^{t,c,c'}$ | The packet size required for transferring VNF instance $m$ on the migration path $l$ between source cloudlet $c$ and target cloudlet $c'$ of IoT request flow $i$ at time slot $t$ |

transformed into a directed graph $G_i = (F_i, L_i)$, where $F_i$ represents the set of VNFs required by the IoT request flow $i$. For any $VNF m \in F_i$, the required computing resource is $f^{cpu}_{i,m}$ and the required memory resource is $f^{mem}_{i,m}$. $L_i$ denotes the set of virtual links of the $SFC_i$. The maximum tolerable end-to-end delay for IoT request flow $i$ is $R^{delay}_i$, the bandwidth resource requirement is $R^{bw}_i$, and the maximum tolerable delay for $SFC_i$ seamless migration is $T_i$.

## Time delay and reconfiguration cost model

**Queue delay.** When data packets from IoT request flow $i$ require processing on a Cloudlet, they typically need to queue in the input buffer. Each Cloudlet within the IoT-MEC network is modeled as an M/M/1 queueing system. The queuing delay experienced by the data packets passing through Cloudlet $c$ during time slot $t$ can be approximated by the average waiting time the packets endure before being processed by the required VNF instance $m$ on that Cloudlet. Therefore, the queueing delay $d_{que}(i, t)$ for IoT request flow $i$ during time slot $t$ can be formulated as follows:

$$d_{que}(i,t) = \sum_{c \in C} \sum_{m \in F_i} x^c_{i,m} \frac{\lambda^t_m}{\mu^t_m(\mu^t_m - \lambda^t_m)}, \forall i \in R \tag{1}$$

Where $\lambda^t_m$ denotes the rate of packet arrivals for VNF $m$ in time slot $t$, which represents the frequency of new packets entering the queue waiting to be processed. $\mu^t_m$ denotes the rate at which Cloudlet $c$ processes packets for VNF $m$ in time slot $t$, showing how fast Cloudlet processes arriving packets. We assumes $\mu^t_m > \lambda^t_m$. The binary variable $x^c_{i,m} \in (0, 1)$, where $x^c_{i,m} = 1$ indicates the request flow $i$ will be processed by the VNF $m$ on Cloudlet $c$, otherwise, the value of $x^c_{i,m}$ is 0.

**Migration delay.** In the MEC network environment, the frequent service migration will greatly increase the additional migration delay of the whole network. Therefore, we intentional formulate the migration model of VNF $m$. We use $Q^{t,c,c'}_{i,m}$ to represent the packet size of the IoT request flow $i$ that needs to be migrated from VNF instance $m$ on the Cloudlet $c$ to the target Cloudlet $c'$ in the time slot $t$, that is, the packet size that has not been processed by the VNF $m$ on the Cloudlet $c$. The migration delay $d_{mig}(i, t)$ of IoT request flow $i$ in the time slot $t$ includes the processing delay and the transmission delay of data packets that need to be migrated to the new Cloudlet $c'$. It is shown in Eq (2):

$$d_{mig}(i,t) = \sum_{c \in C} \sum_{c' \in C} x^{t,c,c'}_i Q^{t,c,c'}_{i,m} \left( \frac{1}{b^{t,c,c'}_{uv}} + \frac{1}{f^t_{c',m}} \right) \tag{2}$$

Where, $x^{t,c,c'}_i$ is a binary variable, $x^{t,c,c'}_i = 1$ shows that the request flow $i$ will migrate from Cloudlet $c$ to Cloudlet $c'$ in the time slot $t$, otherwise, $x^{t,c,c'}_i = 0$ indicates that migration of the request flow $i$ is not performed. $b^{t,c,c'}_{uv}$ indicates that the bandwidth resources usage of physical link $uv$ that the request flow migrate from the Cloudlet $c$ to $c'$. $f^t_{c',m}$ (Hz) represents the processing power of VNF $m$ on the Cloudlet $c'$ in the time slot $t$.

**Transmission delay.** In the time slot $t$, when the data packets of IoT request flow $i$ is transmitted on the physical link, it will generate the transmission delay $d_{trans}(i, t)$, which is defined as follows:

$$d_{trans}(i,t) = \sum_{u,v \in L} \sum_{m,n \in F_i} \frac{Q^t_{i,m}}{b^{t,uv}_{i,mn}} \tag{3}$$

Where $Q_{i,m}^t$ represents the packet amount of arriving at VNF $m$ in the time slot $t$, $b_{i,mn}^{t;uv}$ represents the bandwidth resource on the physical link $uv$ which is mapped by the virtual link $mn$ between adjacent VNF $m$, $n$ of IoT request flow $i$ in the time slot $t$.

**Processing delay.** The processing delay is the time required to process the data packets of IoT request flow $i$ on the corresponding Cloudlet $c$. In the time slot $t$, the processing delay $d_{pro}(i, t)$ on the Cloudlet $c$ is defined as follows:

$$d_{pro}(i, t) = \sum_{c \in C} \sum_{m \in F_i} x_{i,m}^c \frac{Q_{i,m}^t}{f_{c,m}^t} \tag{4}$$

Where $f_{c,m}^t$ (Hz) denotes the processing power of VNF $m$ on Cloudlet $c$ in time slot $t$.

**Total delay.** Therefore, when the IoT request flow $i$ needs to be migrated at the time slot $t$, the total end-to-end time delay is shown in Eq (5):

$$D(i, t) = d_{que}(i, t) + d_{mig}(i, t) + d_{trans}(i, t) + d_{pro}(i, t), \forall i \in R \tag{5}$$

**Reconfiguration cost.** Due to the fact that SFC migration is essentially data migration, we approximate the cost of data migration to the bandwidth overhead generated during SFC rerouting, which is mainly related to the size of data to be transmitted. In the scenario where the VNFs of SFC $i$ are migrated from Cloudlet $c$ to Cloudlet $c'$ within the time slot $t$, the Reconfiguration cost is defined as follows:

$$C(i, t) = \sum_{l \in L} \sum_{c \in C} \sum_{m \in F_i} c_l x_i^{t,c,c'} y_l^{t,c,c'} (Q_{i,m}^{t,c,c'} / 100) \tag{6}$$

Where $c_l$ indicates the transmission cost associated with every 100 Mbit of data traversing through link $l$ [24]. The $y_l^{t,c,c'}$ is a binary variable. When $y_l^{t,c,c'} = 1$, it indicates that the data packets migrated by VNF from Cloudlet $c$ to Cloudlet $c'$ will use the physical link $l$ as its reroute path.

## Problem definition

Given an IoT request flow $i$ with SFC requirements in the IoT-MEC, the objective of our work is to jointly consider the mobility of IoT users or IoT devices, real-time load changes and resource constraints of the edge servers to minimize the reconfiguration cost and end-to-end delay of seamless migrating IoT request flow $i$. This is a multi-objective optimization problem according to the delay model and cost model described above. In order to obtain the optimal solution, the two objectives need to be normalized to the same order of magnitude and weighted. We define the optimization objective as follows:

$$\mathcal{U} = \alpha \frac{D(i, t)}{D_{max}} + \beta \frac{C(i, t)}{C_{max}} \tag{7}$$

Where, $\alpha$ and $\beta$ are weighted coefficients used to balance the end-to-end delay and migration cost, and $\alpha + \beta = 1$. $D_{max}$ represents the maximum tolerable delay of IoT request flow in time slot $t$, and $C_{max}$ is the maximum migration cost of IoT request flow in time slot $t$. Therefore, the optimization problem solved is shown in Eq (8):

$$\min : \mathcal{U} \tag{8}$$

Subsequently, we will provide the constraints of the above optimization problem.

We assume a VNF $m$ can only be deployed on a Cloudlet $c$, the Eq (9) must be satisfied:

$$\sum_{c\in C} x_{i,m}^c = 1, \forall i \in R \tag{9}$$

Eq (10) ensures that the virtual link between adjacent VNF $m$ and $n$ in the SFC should be mapped to a acyclic path in the physical network:

$$\sum_{l\in L} y_{i,mn}^l = 1, \forall i \in R \tag{10}$$

Where, $y_{i,mn}^l$ is a binary variable, $y_{i,mn}^l = 1$ represents the virtual link between VNF $m$ and $n$ of IoT request flow $i$ is maped to physical link $l$.

The end-to-end delay constraint of IoT request flow $i$ is shown in Eq (11):

$$D(i, t) \leq R_i^{delay} \tag{11}$$

To ensure seamless migration of SFC, Eq (12) shall be satisfied:

$$d_{mig}(i, t) \leq T_i \tag{12}$$

To ensure that the CPU resource occupied by the VNF instances on the Cloudlet $c'$ is less than or equal to the available CPU capacity of the Cloudlet $c'$, Eq (13) must be satisfied:

$$\sum_{i\in R} \sum_{m\in F_i} x_{i,m}^{t,c,c'} f_{i,m}^{cpu} \leq C_{c'}^t, \forall t \in T, c' \in C \tag{13}$$

Where, $C_{c'}^t$ represents the available CPU resources at the time slot $t$ in Cloudlet $c'$.

Eq (14) ensures that the bandwidth resource constraint on the reroute path is not violated:

$$\sum_{i\in R} \sum_{m\in F_i} y_{i,mn}^l R_i^{bw} \leq b_l^t, \forall t \in T, l \in L \tag{14}$$

Where, $b_l^t$ represents the available bandwidth resources on link $l$ at the time slot $t$.

After the reconfiguration of the SFC, the Cloudlet to which VNF instance accommodated before and after migration should be different, it is shown in Eq (15):

$$v_{i,m}^t \neq v_{i,m}^{t+1} \tag{15}$$

Where, $v_{i,m}^t$ represents the Cloudlet which the VNF instance $m$ of IoT request flow $i$ is hosted on at the time slot $t$.

To ensure the continuity of reroute path, Eq (16) shall be satisfied:

$$\sum_{d\in C} \sum_{mn\in L_i} \left( y_{i,mn}^{l,s,d} - y_{i,mn}^{l,d,s} \right) = \begin{cases} 1, s \text{ is source node} \\ -1, s \text{ is destionation node} \\ 0 \end{cases} \tag{16}$$

In this paper, due to the multiple paths are used to migrate packets at the same time, the sum of migrated data on all rerouted paths during the migration process should be greater than the total amount of data to be migrated, it is shown in Eq (17):

$$\sum_{l\in P_{i,t}^m} Q_{i,m,l}^{t,c,c'} \geq Q_{i,m}^{t,c,c'} \tag{17}$$

Where, $P_{i,t}^m$ represents the migration path set of VNF instance $m$ of the IoT request flow $i$ in the

time slot $t$. $Q_{i,m,l}^{t,c,c'}$ represents the size of data packets that need to be migrated when transmitting VNF instance $m$ on the migration path $l$ between source Cloudlet $c$ and target Cloudlet $c'$ in the time slot $t$.

The reconfiguration problem of SFC in IoT-MEC network can be reduced into a facility location problem with capacity constraints, which is a NP-hard problem [38]. Therefore, the problem in this paper is also a NP-hard problem. We next propose a prediction based reconfiguration scheme of SFC to solve this NP-hard problem.

## Design of Prediction based SFC Active Reconfiguration (PSAR) algorithm

**Trajectory prediction model for mobile users.** When IoT devices move in the environment, they will frequently change base stations, which may lead to network service interruption, data communication delay increase and the QoS decline of user. Therefore, it is necessary to design a mobility prediction model to accurately predict the user's position in the next time slot, and then realize effective SFC active reconfiguration for users. Usually, mobile intelligent devices such as intelligent wearable devices, drones, autonomous driving, AR, and VR are equipped with positioning function, which collect a large amount of motion trajectory data, reflecting the movement trajectory information of terminal holders, providing data support for mobile user trajectory prediction. The proposed moving trajectory prediction method by us is mainly illustrated as follows:

*Data pretreatment.* We obtain the historical trajectory information of a mobile user at a certain time interval $t$ for a continuous period of time $T_{obs}$. The historical trajectory is composed of the spatial information of geographic coordinate points composed of longitude and latitude, and the time information of the timestamp when it reaches the region. Usually, there is a large amount of error and noise data between the spatiotemporal activity trajectory of mobile terminals and the base station, which can be manifested as: (1) a large amount of duplicate data generated due to users staying in certain locations for a long time. (2) When a mobile user is located at a point where the coverage of two base stations overlaps, it causes the mobile terminal signaling to quickly switch back and forth between the two base stations, which results in ping-pong phenomenon. To effectively obtain the historical trajectory data of mobile terminals, it is necessary to preprocess the historical trajectory information and eliminate these errors and noise data.

We denote the historical trajectory data as $X = (l_1, l_2, \cdots, l_{T_{obs}})$, where $T_{obs}$ represents the length of the observation sequence, $l_t = (lng_t, lat_t)$ represents the trajectory point at the time $t$, $lng_t$ and $lat_t$ respectively represent the longitude and latitude coordinates of the user's position at the time $t$. For dealing with the large amount of duplicate data generated by users' long stay in some places, the adjacent data in the same position in the trajectory sequence are merged, and only the duplicate data that appears for the first time is retained. Additionally, we directly use deletion method to filter and process ping-pong data.

In order to obtain fine-grained prediction results while protecting the user's privacy as much as possible, the trajectory data is preprocessed using a location information gridding method after removing the noise. Specifically, the MEC network will be gridded, with each grid consisting of a regular hexagonal cellular network with a radius of $r$ meters. We use $g_{lng}^j$ and $g_{lat}^j$ to respectively represent the center longitude and latitude coordinates of the $j^{th}$ grid. The calculation of the grid number mapped to each trajectory point in the trajectory sequence

is shown in Eq (18):

$$f_t = \arg\min_{j} \sqrt{\left(lng_t - g^j_{lng}\right)^2 + \left(lat_t - g^j_{lat}\right)^2} \tag{18}$$

After grid processing, the $l_t$ in the trajectory points is mapped to the grid number $f_t$, and then the grid number is converted to the grid index number $idx_t$ using One-Hot Encoding. Furthermore, a sequence of mobile user trajectories represented by grid index numbers can be obtained, that is $X = (idx_1, idx_2, \cdots, idx_{T_{obs}})$.

*Encoder-decoder prediction model based on attention mechanism.* In this paper, we propose a mobile user trajectory prediction model based on an encoder-decoder architecture that decouples feature extraction and prediction. As shown in Fig 2, the model consists of three main components: encoder, decoder, and attention layer. The encoder uses a Bi-LSTM network to extract time series features from historical trajectory data. The decoder consists of an LSTM network and a fully connected layer. The attention layer assigns different weights to the hidden layer vectors based on their importance in predicting the trajectory points at each time step. The Bi-LSTM network processes the input sequences in both positive and negative directions and then merges the output vectors to determine the final hidden layer vectors. In this way, the model is able to take into account the influence of future data on the current time. The attention layer computes the context vectors by assigning different weights to the hidden layer vectors based on their importance in predicting the trajectory points at each time step. The LSTM layer of the decoder receives the prediction result of the previous time, the hidden state and the context vector of the previous time to generate the hidden state vector at that time. Then, it looks up the context vector and hidden state vector of the current time slot to generate the predicted output value of the current time slot. The fully connected neural network processes the output of the hidden layer of the encoder and outputs the probability that a mobile user stays on each grid using a soft-max function. The model is trained using backpropagation and the network parameters are iteratively updated. The cross-entropy loss function is used to measure the difference between the predicted location labels and the true location labels.

Considering that the trajectory data of mobile users has the characteristics of time series data, and the results of time series prediction data will be affected by the future data, therefore, we select Bi-LSTM as the Encoder. The neural network of Bi-LSTM includes two independent LSTMs. The input sequence is input into the two LSTM neural networks in positive order and reverse order respectively for feature extraction, and we then obtain two output vectors. The

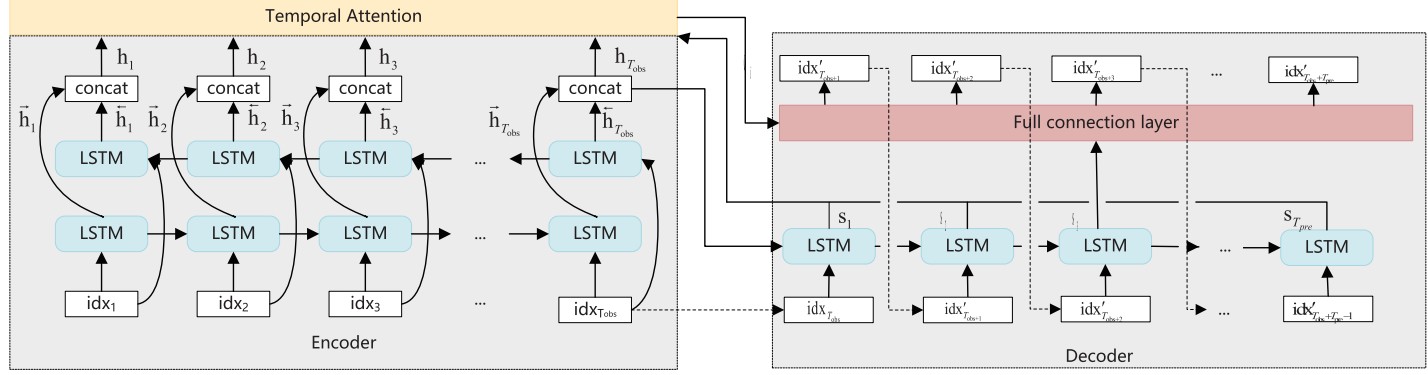

**Fig 2. Encoder-decoder prediction model based on attention mechanism.**

output of Bi-LSTM will be jointly determined by the two output vectors [39]. The Bi-LSTM considers the influence of the future data on the data of the current time. We input the preprocessed grid sequence into Bi-LSTM to extract depth time correlation characters. After $T_{obs}$ time steps of selective memory and forgetting, we concatenate the hidden vector outputs of LSTM in both directions as the final output hidden layer vector $h_i$. The $\overleftarrow{h_i}$, $\overrightarrow{h_i}$ and $h_i$ are calculated by Eqs (19), (20) and (21), respectively.

$$\overleftarrow{h_i} = \overleftarrow{f}(idx_i) \tag{19}$$

$$\overrightarrow{h_i} = \overrightarrow{f}(idx_i) \tag{20}$$

$$h_i = [\overleftarrow{h_i}, \overrightarrow{h_i}] \tag{21}$$

In Eqs (19) and (20), $\overleftarrow{f}$ and $\overrightarrow{f}$ are the LSTM mapping functions of the reverse and forward directions, respectively.

Due to the importance of trajectory points at each time in the trajectory sequence of mobile users is different, nevertheless, the Bi-LSTM does not distinguish these long-time trajectory sequences. Therefore, we input the final hidden layer variable into the attention layer, so that the attention layer can give different weight values to the hidden layer vector. The weight value denotes the importance of the features extracted at different time points. The greater the weight value, the greater the contribution of the features, we then sum the obtained weight value and the hidden layer output as the new output characteristic value of the hidden layer, that is, the context vector $C_i$. The calculation of $C_i$ is shown in Eq (22):

$$C_i = \sum_{j=1}^{T_{obs}} a_{ij} h_j \tag{22}$$

In the above equation, $h_j$ is the $j^{th}$ hidden vector on the Encoder end. The $a_{ij}$ indicates how much attention should be paid to the $j^{th}$ input when outputting the $i^{th}$ predicted value, which is a softmax function output, and $\sum_{j=1}^{T_{obs}} a_{ij} = 1$, the $a_{ij}$ is calculated by the Eq (23):

$$a_{ij} = \frac{\exp(e_{i,j})}{\sum_{k=1}^{T_{obs}} \exp(e_{i,k})} \tag{23}$$

In the Eq (23), $e_{i,j}$ is an alignment model which is calculated by the following Eq (24):

$$e_{i,j} = v^T \tanh(W[h_j; s_{i-1}]) \tag{24}$$

Where $W$ is a weight matrix, $v$ is a weight vector, $s_{i-1}$ is the hidden state of the Decoder in time slot $t$, $s_{i-1}$ is calculated by the following Eq (25):

$$s_i = f_d(idx_{T_{obs}+i-1}, C_i, s_{i-1}) \tag{25}$$

Where $f_d$ is the mapping function of LSTM on the Decoder.

The LSTM in the first layer of the Decoder receives the prediction results of the previous time, the hidden states of the previous time and the context vector $C_i$ to generate the hidden state vector at that time, and then looks up the context vector $C_i$ and the hidden state vector of the current time slot to generate the predictive output value of the current time slot. In the

model, the fully connected neural network processes the output of the hidden layer of the LSTM in the Encoder. In the output layer of the fully connected neural network, the softmax function is used to output the probability that the mobile user stays in each grid, and the grid corresponding to the index with the largest probability is taken as the most likely stay location of the mobile user at the next moment.

We use the back propagation method to train the prediction model, and update the network parameters iteratively. The cross entropy loss function is shown in Eq (26):

$$L = -\frac{1}{T_{pre}} \sum_{i=1}^{T_{pre}} \sum_{j=0}^{N-1} y_{T_{obs}+i,j} \log y'_{T_{obs}+i,j} \qquad (26)$$

where $y_{T_{obs}+i,j}$ represents the true position label in the $T_{obs} + i$ time slot, $y'_{T_{obs}+i,j}$ represents the probability that the model predicts the mobile user is in grid $j$ at the time slot $T_{obs} + i$.

## Node load prediction model

To achieve a more accurate resource demand prediction of SFC, we jointly consider the historical resource demand data of a single VNF and the characteristic information of other VNFs in SFC. In this paper, we assume that the VNF can be shared by multiple SFCs, it is necessary to consider VNF resource usage information in the current SFC and other SFCs as training data. This is because VNF will receive traffic from other VNFs, when the adjacent VNFs of the target VNF have insufficient resources, the garbage traffic may be generated or the connection with the target VNF may be interrupted [40]. Using the resource usage information of other VNFs as training data can greatly improve the resource demand prediction accuracy. Especially, the historical resource usage information of adjacent VNFs is more important for predicting the resource demand of VNF instances compared to other VNFs in the same SFC [40]. As shown in Fig 3, VNF2 is a common VNF for SFC1 and SFC2. To predict the computing or memory resource requirements of VNF2 instances at the next moment, it is necessary to input the resource usage information of VNF1, VNF3, VNF4, and VNF5 into the prediction model to assist in prediction.

In addition, due to the real-time changes in resource requirements of SFC, after a period of time, the initially trained model may make significant errors in predicting new samples. However, retraining the model will consume a significant amount of time cost, therefore, we propose a model that utilizes parameter training to assist in online prediction.

As shown in Fig 4, the VNF instance resource demand prediction model in this paper consists of two parts: parameter training and online prediction. In the parameter training stage, $N$ historical data are first extracted from the experience replay pool using a sliding window. The parameter training model performs forward batch training on the historical data, and then passes the trained model parameters to the online prediction model to improve the training

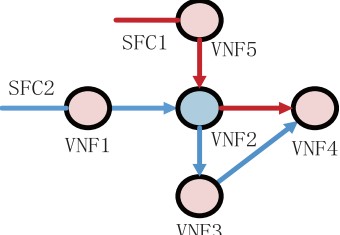

**Fig 3. VNF sharing example in Multi SFC.**

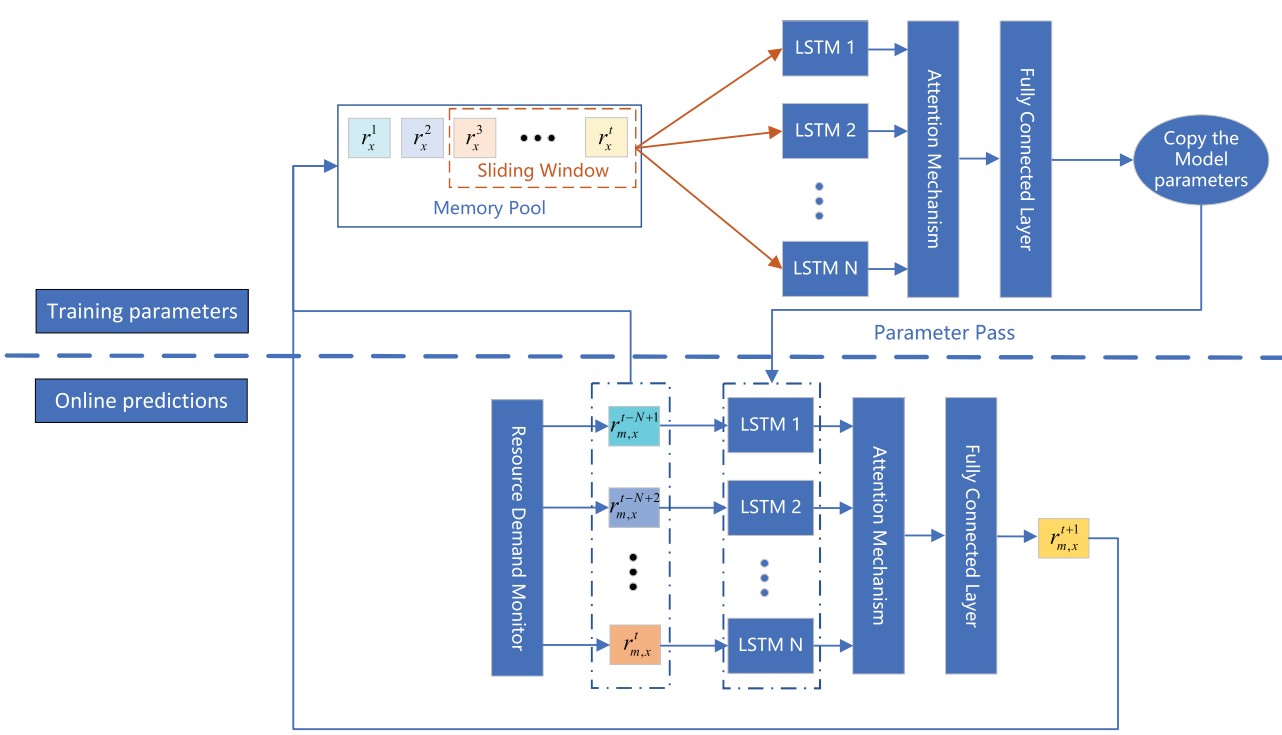

**Fig 4. The resource demand prediction model of VNF instance.**

speed. In the online prediction stage, the real-time optimization of the prediction model is achieved. Due to the real-time changing resource requirements of SFC, after a period of time, the initially trained model may make significant errors in predicting new samples, therefore, we use the results of parameter training to assist the online prediction model. Firstly, we use a demand monitor to collect features of VNF's CPU and memory resource requirements to obtain historical data on VNF's CPU and memory resource usage. The resource usage historical data of VNF $m$ is represented by Eq (27). When $x = cpu$, it represents the CPU usage historical data of VNF $m$. When $x = mem$, it represents the memory resource usage historical data of VNF $m$. In this paper, we train a more appropriate model parameter for different types of resources to assist the online prediction model in predicting resource demand.

$$r_{m,x} = \{r_{m,x}^1, r_{m,x}^2, ..., r_{m,x}^t\} \tag{27}$$

where $r_{m,x}^t$ represents the CPU or memory resource usage historical data of VNF $m$ during the $t$ time slot.

In the prediction stage, we first extract the resource usage history information $r_{m,x}$ associated with the target VNF $m$ observed by the resource demand monitor. It is worth noting that in the historical records of node resource usage, the closer the historical records are to the $t + 1$ time slot, the greater the impact on the prediction results. Hence, we utilize the sliding window to intercept the latter $N$ items of the history information and input them to the LSTM network. Since the resource usage history information of each VNF has different impacts on the prediction of the target VNF resource demand, the information learned by LSTM is input to the attention mechanism to assign different weights to each VNF. The next moment resource demand of the VNF is obtained from the trained multilayer neural network. Finally, we

integrate and add the output of the online prediction model and the information obtained from the resource demand monitor into the memory pool as the training set for the parameter training model.

## Service function chain reconfiguration algorithm

The SDN controller in the MEC network monitors the state of the physical network, estimate the QoS of network at the next moment based on real-time user mobility and VNF instance resource demand prediction results of SFC, and determines whether to initiate the active reconfiguration mechanism of SFC. This section will illustrate in detail how to determine the edge server to be migrated when the VNF's demand for the underlying physical network resources exceeds the resource remaining capacity of the node or the end-to-end delay constraint of SFC is violated due to user's movement, and how to allocate the reroute path for SFC when the VNF's migration is accomplished.

Algorithm 1 shows the SFC reconfiguration trigger algorithm. The trigger condition of the algorithm is that the end-to-end delay constraint of SFC is violated due to user's movement or the resource demand of VNF instance exceeds the node load range. In line 2, the set $N_i$ stores the overloaded nodes in SFC $i$, the set $N$ stores the all overloaded nodes in the network. In lines 3-8, the algorithm predict the resource requirements of each VNF $m$ in SFC $i$, and calculate the CPU resource utilization and memory resource utilization of physical node based on the predicted result. If the predicted result is greater than the upper resource utilization threshold or less than the lower resource utilization threshold, the nodes corresponding to VNF $m$ of SFC $i$ will be added to the overloaded node set $N_i$. In lines 9-12, the algorithm adds the SFC that triggers SFC reconfiguration condition due to VNF resource requirements exceeding the node resource utilization range to the set $S_1$, and simultaneously, adds the set $N_i$ to the set $N$. In lines 14-16, the algorithm adds the SFC request flows that violate end-to-end delay constraints to the set $S_1$ and the set $S_2$, as a result, the SFCs that only violate the end-to-end delay constraints but does not have overload nodes are stored in the set $S_2$, and the SFCs with VNF instance resource requirements exceed the node resource utilization range but meeting the end-to-end delay requirements are stored in the difference set between $S_1$ and $S_2$.

**Theorem 1** *The time-complexity of Algorithm 1 SFC Reconfiguration Trigger Algorithm is $R^*|F_i|$.*

**Proof 1** *The time complexity of this algorithm mainly depends on the number of nodes $N_i$, the number of VNF instances $|F_i|$. The for loop in line 1 of the algorithm will run a maximum of $R$ times. The initialization process line 2 of the algorithm has a time complexity of $O(1)$. For each node VNF $m$ in $F_i$, the algorithm requires traversing all nodes corresponding to VNF in $F_i$ to identify the node that violates the resources constraint, this has a time complexity of $(|F_i|)$. The trained prediction model is then invoked for each VNF instance with a time complexity of $O(1)$. Finally, the time complexity of calling the trained Encoder-Decoder model in line 13 of the algorithm for prediction is $O(1)$. Combining the above steps, the total time complexity is $R^*|F_i|$.*

After obtaining the SFCs that need to be reconfigured, we should determine the VNFs to be migrated out and the edge servers to be migrated in for achieving seamless migration. Algorithm 2 shows the VNFs migration process. In lines 3-14, the algorithm perform VNFs migration for SFCs that trigger reconfiguration only due to not satisfied end-to-end latency. First, the migration order of VNFs in an SFC is sorted according to the resource utilization of its corresponding physical nodes, that is, the VNF on which the node has the highest load will be migrated first. For each VNF $m$, lines 6-7 of the algorithm look for candidate migrating nodes that meet the conditions of resource constraints, seamless migration, and end-to-end delay and store them in the set $C$, then, the algorithm selects the node with the lowest migration cost

and end-to-end delay from the set $C$ as the node that the VNF $m$ will migrate in, the algorithm then calls Algorithm 3 to obtain the reroute path of SFC. Lines 15-20 of the algorithm migrate all VNF instances in SFC whose resource requirements exceed the load range of the corresponding physical node.

**Algorithm 1** SFC Reconfiguration Trigger Algorithm

**Require**: SFC request set: $R$.
**Ensure**: SFC set that needs to be reconfigured: $S_1$.
1: **for** each SFC $i$ in $R$ **do**
2:     Initialize node set $N_i$, Overload node set $N$, SFC set $S_1$ that needs to be reconfigured, the SFC set $S_2$ that violates end-to-end latency, they are all initialized as empty sets.
3:     **for** each VNF $m$ in $F_i$ **do**
4:         Use the trained load prediction model to predict the memory resource demand $r_{m,mem}^{t+1}$ and CPU computing resource demand $r_{m,cpu}^{t+1}$ of VNF instance $m$ in the next time slot. Based on the prediction results, calculate the CPU resource utilization rate $\mu_{cpu}$ and memory resource utilization rate $\mu_{mem}$ of the physical node.
5:         **if** $\left(\mu_{cpu} > \widehat{\mu}_c \cup \mu_{cpu} < \breve{\mu}_c\right) \cup \left(\mu_{mem} > \widehat{\mu}_m \cup \mu_{mem} < \breve{\mu}_m\right)$ **then**
6:             Add VNF $m$ to $N_i$.
7:         **end if**
8:     **end for**
9:     **if** $N_i \neq \emptyset$ **then**
10:         Add $N_i$ to $N$.
11:         Add SFC $i$ to $S_1$.
12:     **end if**
13:     Call the trained Encoder-Decoder movement trajectory prediction model based on attention mechanism to predict the user's position in the next time slot, and calculate the end-to-end delay $D(i, t + 1)$ of $SFC_i$ in the next time slot based on the prediction results.
14:     **if** $D(i, t + 1) > D_i$ **then**
15:         Add SFC $i$ to $S_1$ and $S_2$, respectively.
16:     **end if**
17: **end if**

Due to the migration of SFC can be regarded as the migration of the unprocessed data on the VNF. For the given source edge node and target edge node to be migrated in, we can use multiple routing paths to migrate data packets at the same time so as to efficiently transmit data. Algorithm 3 describes the selection of reroute path in detail, in which the line 5 of the algorithm obtains the minimum bandwidth value of all links on the shortest path $p$.

**Algorithm 2** VNF Migration Algorithm

1: Initialize the edge node set $E$ to be migrated in as empty
2: **for** each SFC $i$ in $S_1$ **do**
3:     **if** $N_i = \emptyset$ **then**
4:         Sort VNF $m$ in $F_i$ by resource utilization on its nodes.
5:         **for** VNF $m$ in $F_i$ **do**
6:             Select candidate nodes that meet resource constraints, seamless migration, and end-to-end latency and store them in the set $C$.
7:             Find the node with the smallest migration cost and end-to-end delay in $C$ as the node $n$ to be migrated in.
8:             Add $n$ to $E$.
9:             Call Algorithm 3 to obtain the rerouting path.
10:             **if** $D(i, t) \leq R_i^{delay}$ **then**
11:                 break
12:             **end if**

```
13:      end for
14:    else
15:      for VNF m in N_i do
16:        Select candidate nodes that meet resource constraints, seam-
               less migration, and end-to-end latency and store them in the
               set C.
17:        Find the node with the smallest migration cost and end-to-end
               delay in C as the node n to be migrated in.
18:        Add n to E.
19:        Call Algorithm 3 to obtain the rerouting path.
20:      end for
21:    end if
22: end for
```

**Theorem 2** *The time-complexity of the Algorithm 2 VNF Migration Algorithm is $O(R|F_i|N)$.*

**Proof 2** *The time complexity of this algorithm mainly depends on the number of nodes N, the number of VNF instances $|F_i|$ and the size of the edge node set E. For the resource utilization operation of each node, the VNF instances of each node need to be sorted with a time complexity of $O(|F_i| \log |F_i|)$ (line 4). The operation of selecting candidate nodes and calculating the migration cost requires traversing all nodes and VNF instances with a time complexity of $O(N|F_i|)$ (lines 6-8, lines 16-18). Algorithm 3 is invoked to obtain the operation of rerouting paths, and the time complexity of Algorithm 3 is usually $O(|F_i|N)$ (line 9, line 19). The operations of updating node states and migrating VNF instances require traversing all VNF instances with a time complexity of $O(|F_i|)$. Combining the above steps, the overall time complexity of the whole algorithm can be expressed as $O(S_1*(|F_i| \log |F_i| + N|F_i| + |F_i| + |F_i|N))$, in the worst case, $S_1 = R$, therefore, the time complexity of the Algorithm 2 is $O(R|F_i|N)$.*

**Algorithm 3** Rerouting path selection Algorithm

```
Require: VNF m to be migrated; Edge node n to be migrated in.
Ensure: Reroute path set P.
1: Initialize migrated packet size: Q_{i,m} = 0.
2: Construct a weighted graph G' of the original edge network G, where
   the edge node to be migrated is divided into two virtual nodes. The
   edge weight between the two virtual nodes is the processing delay
   which is computed by Eq (4), and the weight of each other edge in
   the G' is the weighted sum of reconfiguration cost which is computed
   by Eq (6) and the transmission delay of physical link which is com-
   puted by Eq (3) in the G.
3: while Q_{i,m}^{t,c,c'} ≥ Q_{i,m} do
4:    Use Dijkstra algorithm to obtain the shortest path p between the
      node corresponding VNF m and the node n.
5:    B_{mn}^p = min (B_l)_{l∈p}.
6:    Add p to P.
7:    Q_{i,m} = Q_{i,m} + T_i * B_{mn}^p.
8:    Update the weights of edges in G'.
9: end while
```

## Performance evaluation

### Simulation settings

**Dataset.** We use a public data set Geolife [41] provided by Microsoft Research Asia to train the trajectory prediction model of mobile users. This public data set contains trajectory records of 182 users collected through GPS between April 2007 and August 2012, including locations in over 30 cities in China and several cities in the United States and Europe. In this paper, we only use the tracks from Beijing city, we select 270 user trajectory records from the

suburbs of 2000 m * 2000 m, and use only their longitude, latitude, and times tamp information for each trajectory record. The trajectory sampling time interval is 30 seconds. We randomly select 80% of the data as the training set and the remaining 20% of the data as the test set, our trajectory prediction model achieves a 95% accuracy of rate. Our load prediction model is evaluated on the real data set Materna Traces (BitBrain) [42]. This data set collects the data of 1750 virtual machines in the distributed cloud data center within three months. The data contains 12 characteristic indicators of virtual machines, including CPU utilization, memory utilization, etc. We also randomly selected 80% of the data as the training set and the remaining 20% of the data as the test set. According to the test results, the prediction accuracy of the node load prediction model proposed in this paper reaches 96%.

**Network topology.** We construct a MEC network consisting of 150 MEC servers. The radius of the hexagonal cellular network is 200 m. The CPU resources of each edge server are set from 20 GHz to 25 GHz, the memory capacity from 30 GB to 40 GB, and the bandwidth capacity of each link from 500 Mbps to 1,000 Mbps. In particular, to explore the potential of network topology changes on the effectiveness of the algorithm, 0 to 10 MEC servers are randomly selected to set the CPU resources and memory capacity set to 0 to mimic the impact on the algorithm in case of node failure. The transmission cost $c_l$ is from 10 to 30. $\breve{\mu}_c$ and $\breve{\mu}_m$ are set to 0.2, $\breve{\mu}_c$ and $\breve{\mu}_m$ are set to 0.8 [15], respectively.

**SFC.** Each SFC consists of 3 to 6 VNFs. The maximum tolerable end-to-end delay $D_i$ is set from 50 ms to 100 ms, the delay threshold $T_i$ for seamless migration is set to 15 ms, and the bandwidth resource requirement is set from 20 Mbps to 120 Mbps. The CPU resource requirement of each VNF is from 1 GHz to 3 GHz, and the memory resource requirement is from 2 GB to 4 GB.

**Comparison algorithm.**

- DPSM [24]: In the DPSM algorithm, migration is triggered when the SFC delay caused by a user's movement across base stations is not satisfied to support seamless migration of mobile user services.

- DDQ [15]: DDQ predicts the migration cost of SFC and the bandwidth cost of link in the network based on GNN, and then uses a deep Dyna-Q (DDQ)-based approach to complete SFC reconfiguration, the algorithm doesn't consider user mobility and SFC delay.

- TSRFCM [43]: TSRFCM introduces the fuzzy C-means algorithm based on the original tabu algorithm to obtain the optimal SFC reconfiguration strategy, the TSRFCM method mainly considers network resource utilization and ignores SFC reconfiguration time.

- OSA [44]: OSA reschedules the placement of VNF using the fundamentals of optimal stopping theory, the algorithm considers the latency of SFC but does not consider user mobility.

## Analysis of simulation results

Fig 5 demonstrates the curves of the accuracy and loss value of the proposed location prediction module algorithm with the number of training rounds. As shown in the figure, with the increase of the number of training rounds, the loss value gradually decreases until convergence, and the accuracy rate gradually increases, finally reaching about 95% prediction accuracy.

Fig 6 shows the average SFC migration delay of the PSAR algorithm and the comparison algorithm proposed in this paper when the number of SFCs changes. It can be seen from the figure that the average migration delay of PSAR and other comparison algorithms increase

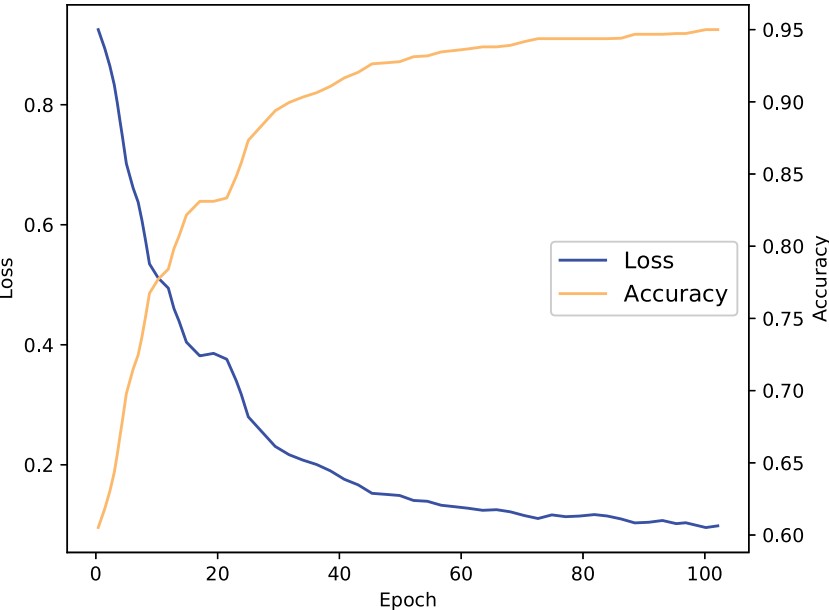

**Fig 5. Loss function and accuracy during training.**

with the increase of the number of SFCs. Although DPSM takes user mobility into account, it assumes that the user's movement trajectory is known and cannot handle the dynamic changes of users, therefore, its average migration delay is higher than PSAR, but lower than the other three algorithms. Owing to the DDQ and TSRFCM algorithms do not consider SFC migration delay, the migration delay performance of the two algorithms are worse than OSA.

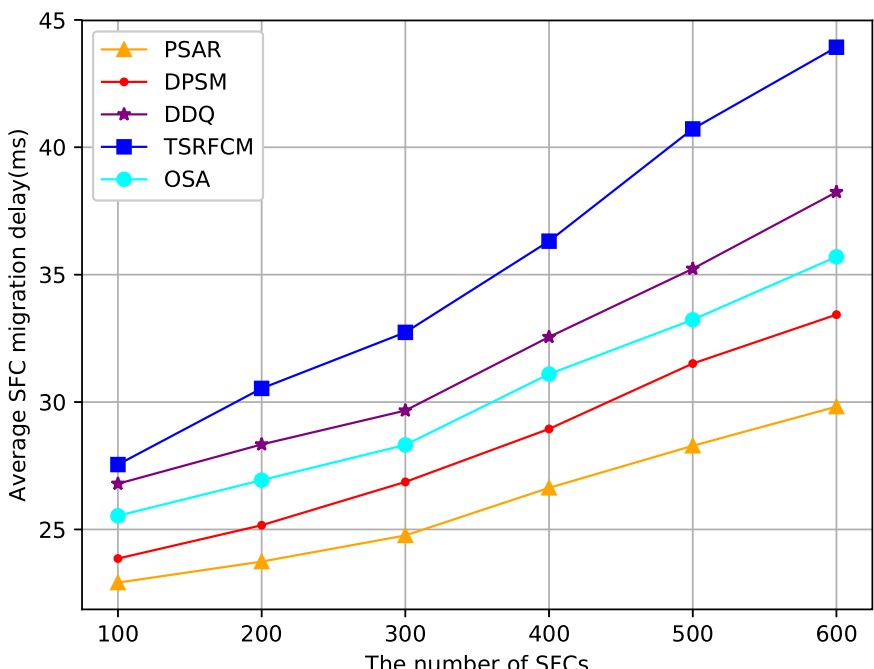

**Fig 6. The variation of average SFC migration delay with the number of SFCs.**

In the Fig 7, the reconfiguration costs of different algorithms when the number of SFCs changes are compared. It can be seen that the reconfiguration costs of all algorithms are almost proportional to the number of SFCs. The reason is that the larger the number of SFCs, the higher the probability that the delay constraints are not satisfied due to the high mobility of users, the greater the possibility of resource overload of nodes, which is easy to trigger SFC reconfiguration, this results in higher and higher reconfiguration costs. In addition, the proposed PSAR algorithm migrates SFC based on user mobility and VNF resource demand prediction, it can proactively migrate SFCs that violate resource and delay constraints in advance, therefore, it's reconfiguration cost is the lowest. The DDQ uses GNN to predict VNF resource demand, which can ensure that VNF instances are migrated to nodes with high resource availability in advance, consequently, the migration cost of DDQ is lower than that of the other three comparison algorithms.

In Fig 8, we compare the average resource utilization rate of the edge server when the number of SFCs changes. The average resource utilization rate is the weighted sum of memory and CPU resource utilization rates on all activated servers compared to the number of activated servers. It can be seen that when the number of SFCs is small, the resource utilization gap between PSAR and other comparison algorithms is small. We can see that the PSAR has the highest resource utilization rate of physical nodes, this is due to the PSAR can not only effectively reserve resources for the future traffic through node load prediction and user movement trajectory prediction, but also it can trigger SFC reconfiguration mechanism for servers with low and high load, therefore, it improves resource utilization rate of edge servers.

Fig 9 illustrates the comparison of average service throughput. When the number of incoming SFCs is 600, the throughput of PSAR is about 5 Gbps higher than that of DDQ, 9 Gbps

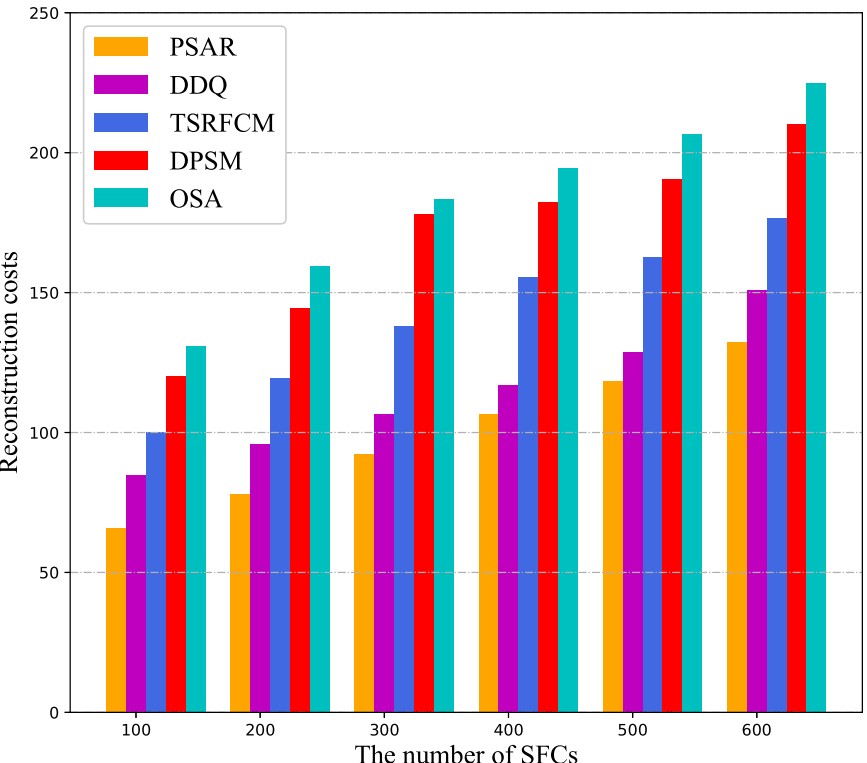

**Fig 7. Variation of reconfiguration cost with the number of SFCs.**

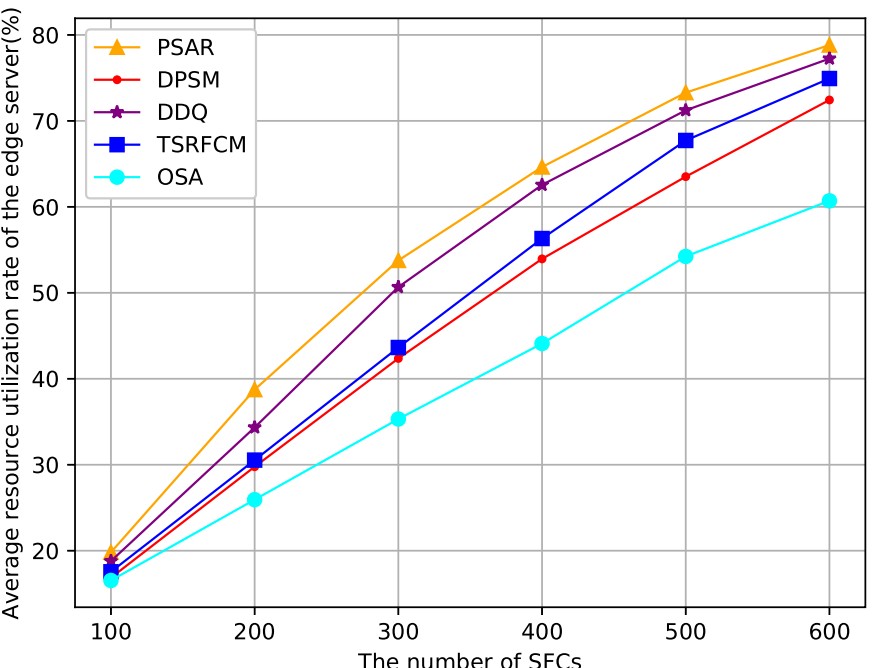

**Fig 8. Average resource utilization rate of the edge server.**

higher than that of DPSM, 10 Gbps higher than that of TSRFCM and 22 Gbps higher than that of OSA. The performance of OSA and DDQ is poorer than that of PSAR and DPSM due to the fact that they don't take into account the mobility of users. Furthermore, PASR algorithm considers the load of nodes and makes the node load more balanced, therefore the throughput performance is higher than other algorithms.

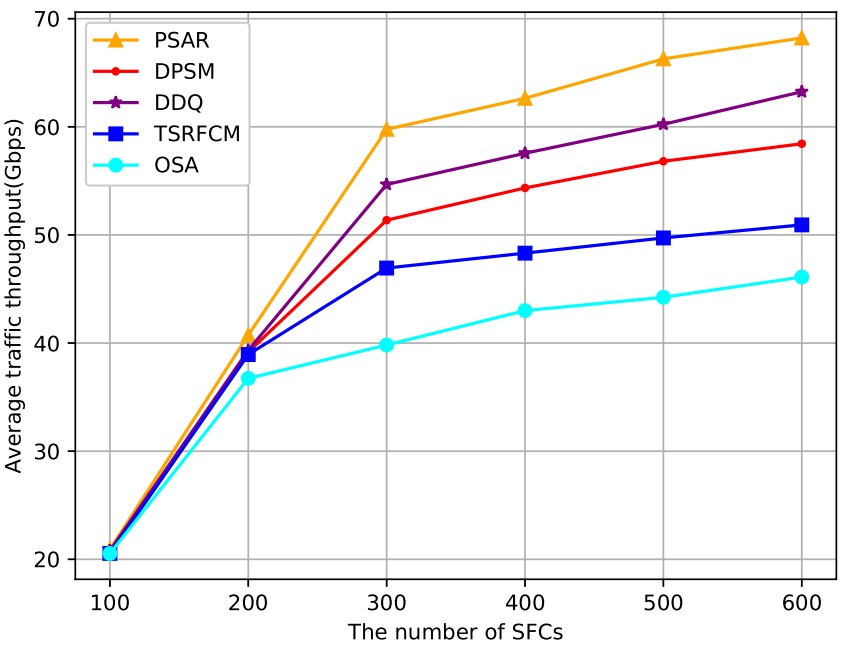

**Fig 9. Comparison of average traffic throughput.**

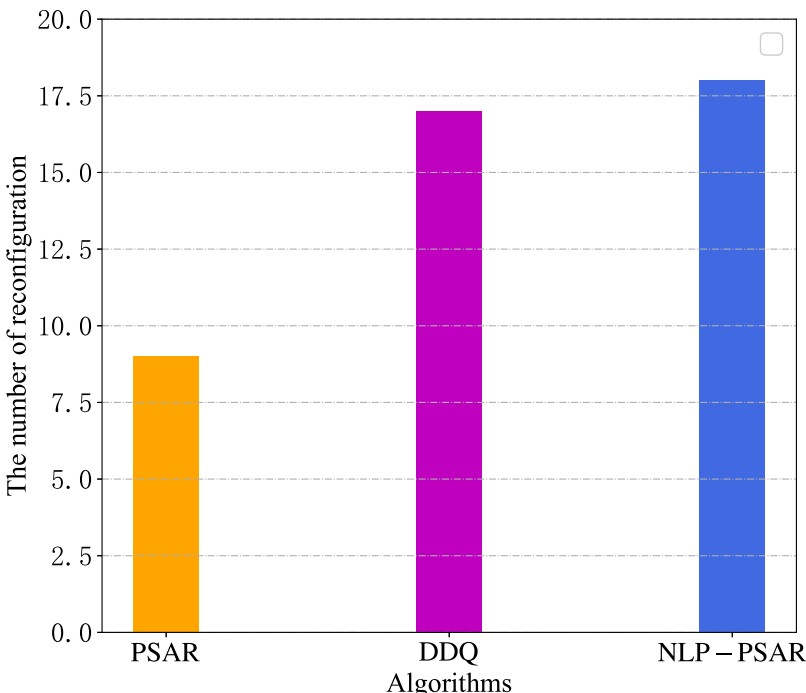

**Fig 10. Comparison of reconfiguration times.**

To evaluate the impact of the node load prediction and user movement prediction on SFC migration performance, we remove the node load prediction section from the PSAR and transform it into a No-Load-Prediction PSAR(NLP-PSAR). Then, we compared the number of SFC reconfiguration with the DDQ algorithm which based solely on node load prediction when the number of SFCs is 50. The results are shown in Fig 10. The PSAR estimates the QoS of the network in the next moment according to the results of user movement trajectory prediction and VNF instance resource demand prediction, and then it determines whether to start the SFC reconfiguration mechanism, therefore, the number of SFC reconfiguration is relatively less. The migration trigger condition of DDQ is that the node load is out of range or violates the delay constraint, it causes less SFC migration than NLP-PSAR.

Fig 11 shows the influence of weight coefficient on average SFC delay and reconfiguration cost of PSAR algorithm. When the value of the weight coefficient $\alpha$ is larger, the PSAR algorithm pays more attention to the average SFC delay. Therefore, as shown in the figure, the average SFC delay decreases with the increase of the weight coefficient $\alpha$, while the SFC reconfiguration cost increases with the increase of $\alpha$.

Fig 12 shows a comparison of the running time of these algorithms. Due to the same time complexity of DDQ and TSRFCM, and the same time complexity of OSA and PSAR, we only compare the running time of PSAR, DDQ, and DPSM. From Fig 12, it can be seen that in the same conditions, as the number of the Cloudlet increases, the running time of all algorithms increases, and the difference gap between them is more significant. The running time trend of these algorithms is in line with their algorithm time complexity.

## Conclusions

In this paper, we investigate proactive reconfiguration strategies for SFCs in dynamic IoT MEC network scenarios induced by high user mobility and real-time changes in network

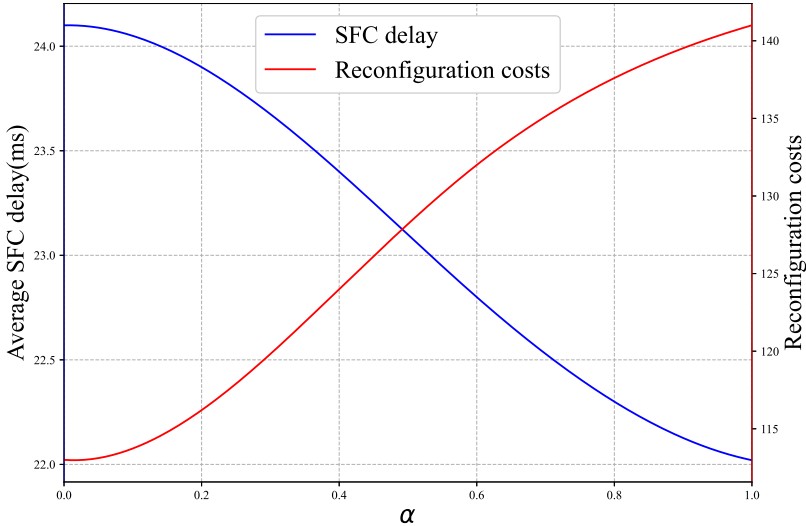

**Fig 11. Performance of PSAR algorithm with different weight coefficient $\alpha$.**

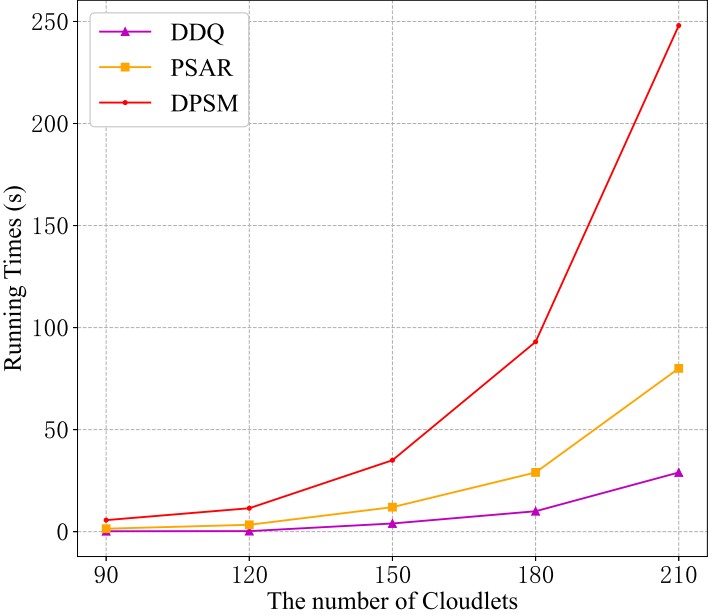

**Fig 12. Running time comparison of different algorithms.**

traffic. First, the SFC reconfiguration problem in the network is defined as the problem of minimizing the end-to-end delay and reconfiguration cost of the SFC and is formulated as an ILP model. Second, an encoder-decoder model based on an attention mechanism is used to predict the user's movement trajectory, and a VNF instance resource demand prediction model is designed to predict the node load. Finally, the QoS of the network in the next time slot is estimated based on the individual predictions obtained from the two prediction models, and a heuristic algorithm, PSAR, is proposed to realize the active reconfiguration of SFC. Simulation results show that the proposed PSAR algorithm outperforms the existing algorithms in terms

of end-to-end reconfiguration performance and effectively improves the reconfiguration efficiency of the system and reduces the end-to-end delay and reconfiguration cost. In future work, since the use of an attention mechanism-based encoder-decoder model to predict the user's movement trajectory in dynamic IoT-MEC network scenarios may need to rely on a large amount of data, the introduction of meta-learning is considered to improve the convergence speed as well as robustness of the model.

## Author Contributions

**Conceptualization:** Shuang Guo, Tengxiang Jing.

**Data curation:** Shuang Guo, Huan Liu.

**Formal analysis:** Shuang Guo.

**Funding acquisition:** Shuang Guo, Huan Liu.

**Investigation:** Shuang Guo, Tengxiang Jing.

**Methodology:** Shuang Guo, Tengxiang Jing.

**Project administration:** Shuang Guo, Liang Liu.

**Resources:** Shuang Guo.

**Software:** Tengxiang Jing, Huan Liu.

**Supervision:** Shuang Guo, Liang Liu.

**Validation:** Tengxiang Jing, Huan Liu.

**Visualization:** Liang Liu, Huan Liu.

**Writing – original draft:** Shuang Guo, Tengxiang Jing.

**Writing – review & editing:** Liang Liu.

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
