## [Decision Letter · Decision Letter 0]

19 Feb 2024

PONE-D-24-02486SFC Active Reconfiguration based on User Mobility and Resource Demand Prediction in Dynamic IoT-MEC NetworksPLOS ONE

Dear Dr. Liu,

Thank you for submitting your manuscript to PLOS ONE. After careful consideration, we feel that it has merit but does not fully meet PLOS ONE’s publication criteria as it currently stands. Therefore, we invite you to submit a revised version of the manuscript that addresses the points raised during the review process.

We look forward to receiving your revised manuscript.

Kind regards,

Praveen Kumar Donta, Ph.D.

Academic Editor

PLOS ONE

Journal Requirements:

"This research was funded by the Science and Technology Research Youth Project of Chongqing Municipal Education Commission under Grant KJQN202202401, Chongqing University of Posts and Telecommunications Doctoral Initiation Foundation under Grant A2023007, Joint Medical Research Program of Chongqing Municipal Science and Technology Bureau and Chongqing Municipal Health Commission under Grant"

"This research was funded by the Science and Technology Research Youth Project of Chongqing Municipal Education Commission under Grant number KJQN202202401, Chongqing University of Posts and Telecommunications Doctoral Initiation Foundation under Grant number A2023007, Joint Medical Research Program of Chongqing Municipal Science and Technology Bureau and Chongqing Municipal Health Commission under Grant number 2024QNXM010."

Please include your amended statements within your cover letter; we will change the online submission form on your behalf. "

 "This research was funded by the Science and Technology Research Youth Project of Chongqing Municipal Education Commission under Grant number KJQN202202401, Chongqing University of Posts and Telecommunications Doctoral Initiation Foundation under Grant number A2023007, Joint Medical Research Program of Chongqing Municipal Science and Technology Bureau and Chongqing Municipal Health Commission under Grant number 2024QNXM010."             

5. For studies involving third-party data, we encourage authors to share any data specific to their analyses that they can legally distribute. PLOS recognizes, however, that authors may be using third-party data they do not have the rights to share. When third-party data cannot be publicly shared, authors must provide all information necessary for interested researchers to apply to gain access to the data. (https://journals.plos.org/plosone/s/data-availability#loc-acceptable-data-access-restrictions)

4) All necessary contact information others would need to apply to gain access to the data.

Reviewers' comments:

Reviewer's Responses to Questions

**Comments to the Author**

1. Is the manuscript technically sound, and do the data support the conclusions?

Reviewer #1: Yes

Reviewer #2: Yes

Reviewer #3: Yes

Reviewer #4: Partly

Reviewer #5: Partly

2. Has the statistical analysis been performed appropriately and rigorously? 

Reviewer #1: Yes

Reviewer #2: Yes

Reviewer #3: Yes

Reviewer #4: N/A

Reviewer #5: No

3. Have the authors made all data underlying the findings in their manuscript fully available?

Reviewer #1: Yes

Reviewer #2: Yes

Reviewer #3: Yes

Reviewer #4: Yes

Reviewer #5: No

4. Is the manuscript presented in an intelligible fashion and written in standard English?

Reviewer #1: Yes

Reviewer #2: Yes

Reviewer #3: Yes

Reviewer #4: No

Reviewer #5: No

5. Review Comments to the Author

Reviewer #1: Clarify the specific performance metrics used to evaluate the SFC-RS, such as throughput, packet loss, or jitter, to provide a more comprehensive assessment of its effectiveness.

Provide a detailed overview of the existing algorithms or methodologies used for SFC reconfiguration in IoT-MEC networks to better contextualize the proposed PSAR algorithm.

Include a discussion on potential limitations or challenges associated with the proposed Encoder-Decoder mobile user trajectory prediction model and the VNF instance resource demand prediction model to address any uncertainties regarding their accuracy and reliability.

Consider integrating additional machine learning techniques or algorithms to further enhance the accuracy of user trajectory prediction and node load estimation, potentially improving the overall performance of the PSAR algorithm.

Discuss the scalability of the proposed SFC reconfiguration strategy in terms of handling a large number of mobile IoT devices and VNF instances, particularly in scenarios with dynamically changing network conditions.

Explore the potential impact of network topology changes, such as node failures or additions, on the effectiveness of the PSAR algorithm and investigate strategies for adaptability and resilience in such scenarios.

Provide insights into the computational complexity of the proposed ILP model and evaluate its scalability with increasing network size and traffic volume to ensure feasibility for real-world deployment.

Consider conducting sensitivity analyses or scenario-based simulations to assess the robustness of the PSAR algorithm under various network conditions and user mobility patterns.

Address potential privacy concerns associated with the collection and processing of user trajectory data and VNF instance resource demand information, and propose measures to safeguard user privacy while maintaining prediction accuracy.

Discuss potential extensions or future research directions for enhancing the SFC reconfiguration strategy, such as incorporating edge computing capabilities or integrating dynamic service chaining mechanisms to adaptively respond to evolving network requirements.

Please avoid citing sources that were published before to 2019. Cite current research that are really pertinent to your topic. The study also lacks sufficient citations. Another critical step is to compare the topic of the article to other relevant recent publications or works in order to widen the research's repercussions beyond the issue. Authors can use and depend on these essential works while addressing the topic of their paper and current issues.

A. Heidari, N. J. Navimipour, M. A. J. Jamali, and S. Akbarpour, "A hybrid approach for latency and battery lifetime optimization in IoT devices through offloading and CNN learning," Sustainable Computing: Informatics and Systems, vol. 39, p. 100899, 2023/09/01/ 2023, doi: https://doi.org/10.1016/j.suscom.2023.100899.

Amiri, Z., Heidari, A., Navimipour, N.J. et al. Adventures in data analysis: a systematic review of Deep Learning techniques for pattern recognition in cyber-physical-social systems. Multimed Tools Appl (2023). https://doi.org/10.1007/s11042-023-16382-x

A. Heidari, N. J. Navimipour, M. A. J. Jamali, and S. Akbarpour, "A green, secure, and deep intelligent method for dynamic IoT-edge-cloud offloading scenarios," Sustainable Computing: Informatics and Systems, vol. 38, p. 100859, 2023.

A. Heidari, M. A. J. Jamali, N. J. Navimipour and S. Akbarpour, "A QoS-Aware Technique for Computation Offloading in IoT-Edge Platforms Using a Convolutional Neural Network and Markov Decision Process," in IT Professional, vol. 25, no. 1, pp. 24-39, Jan.-Feb. 2023, doi: 10.1109/MITP.2022.3217886.

Reviewer #2: This paper proposes a novel SFC adjustment method, and simulation results verify the effectiveness of the method. The reviewer has the following comments and suggestions that might help the authors to improve their work.

->The title of the paper is resource prediction, without specifying what specific resources are, node computing resources, memory resources, or link bandwidth resources.

->Generally speaking, SFC adjustments are made based on deployment, and the deployment mechanism often constrains the adjustment mechanism. How to consider the cost of some adjustment mechanisms being higher than the cost of redeployment?

->The paper mainly considers the triggering factors of load balancing, and is it compatible with indicators such as latency?

->In the literature analysis of the paper, the comparison algorithm did not consider IoT user mobility. However, does the author's use of it as a comparison affect the fairness of the comparison?

->During the experiment, there was a lack of validation of the effectiveness of the predictive model.

->It is suggested to refer to the related work, such as

Forecast-Assisted Service Function Chain Dynamic Deployment for SDN/NFV-Enabled Cloud Management Systems, IEEE SY-J,2023

SFC Embedding Meets Machine Learning: Deep Reinforcement Learning Approaches, IEEE COMML, 2021

Service Function Chain Embedding Meets Machine Learning: Deep Reinforcement Learning Approach, IEEE TNSM,2024.

Reviewer #3: 1.Some existing works have studied similar problem in edge computing by proposing similar methods, what are main differences between the proposed algorithm and existing works.

2.There are a number of grammar errors and typos in this manuscript. The authors have to carefully revise this manuscript before submission. For example,

“The simulation results show that, Compared with … ”

3.Check to use uniform expression formats in context, such as equation(x) and equation x.

4.Please improve the experimental parameter settings and algorithm analysis to demonstrate the superiority of the algorithm in practical use.

Reviewer #4: The paper addresses the problem of live migration of Service Function Chains in distributed edge environments when considering the (i) mobility of users and (ii) the variability of the network conditions. The authors propose a method in which they first predict the user's trajectory and the resource's workload of the computation nodes, then, based on the predicted values, they try to allocate the SFC meaning to minimize the end-to-end delay and the reconfiguration costs. The system was modeled as a multiobjective optimization problem. Finally, the components mentioned were combined into the PSAR algorithm proposed. The PSAR stands for Prediction-based SFC Active Reconfiguration. The authors evaluated the PSAR algorithm through simulations and compared their technique with other online resource allocation algorithms.

The methodology is technically sound and utilizes state-of-the-art solutions to address the problem. The problem that the paper intends to solve is a real issue in current literature, and the approach adopted is creative. Overall, the paper deserves to be published if the authors improve the presentation problems and provide more experiments.

In detail, the paper has two major issues:

1. It has innumerable syntax and grammar mistakes, which are unacceptable in scientific writing. Further, some sentences were written convolutedly, hindering their understandability.

2. The paper needs a broader evaluation of the proposed methods.

Following, I will support how these two issues impact the paper.

To illustrate, I will exemplify some mistakes found only in the paper's abstract. However, the entire paper must undergo careful revision.

"In order to achieve the purpose of secure, reliable and scalable transmission," - > To achieve the purpose of secure, reliable, and scalable transmission,

"Internet of things (IoT)" -> Internet of Things (IoT)

"Due to the high mobility of users and the real-time variability of network traffic in IoT-MEC network, this will cause the mismatch between the performance requirements and the allocated resources of the current SFC." -> Due to the high mobility of users and the real-time variability of network traffic in the IoT-MEC network, this will cause a mismatch between the performance requirements and the allocated resources of the current SFC.

", how to actively reconfigure the deployed SFC according to the changes of network status to ensure the high service quality of the network is a great challenge" -> ; how to actively reconfigure the deployed SFC according to the changes in network status to ensure the high service quality of the network is a great challenge.

"The simulation results show that, Compared" -> lower-case compared

"Prediction based SFC Active Reconfiguration (PSAR)" -> Prediction-based SFV Active Reconfiguration (PSAR)

"based on attention mechanism" based on the attention mechanism

"on Long Short-Term Memory (LSTM)" -> on the Long Short-Term Memory (LSTM)

"aiming at minimizing the end-to-end delay and reconfiguration cost of SFC." -> aiming to minimize the end-to-end delay and reconfiguration cost of SFC.

Regarding the Introduction, avoid non-scientific terms (e.g., more and more and, etc.) The paper begins listing a huge number of initials, which is confusing for the reader. Improve the writing to introduce them more logically.

Two large paragraphs discuss related works in the Introduction. This is before the authors introduce their contributions. Please summarize that part and make it more objective (maybe move part of the content to the related work section).

designs an Predict -> a Predict

"" We utilize PSAR scheme in advance to complete the SFC migration and routing update before the network QoS degrading, and ensure to provide users with consistent high-quality network services"" -> We utilize PSAR scheme in advance to complete the SFC migration and routing update before the network QoS degrades and ensure that users receive consistent, high-quality network services.

Add the outline of the remainder of the work at the end of the Introduction.

Regarding the system model:

The Queue Delay is based on Little's Law, where the λ is the arrival rate and the μ service time. State that more clearly.

The time slot t was never presented, only T.

The total delay is hidden in the processing delay subsection.

A very similar symbol represents the cost objective function. In my first reading, I thought (wrongly) that equation (8) meant that only the cost was minimized. Change one of those symbols to another.

In the Design of PSAR:

We obtain historical trajectory information of mobile users over a continuous period of time at certain time intervals.-> Which period and which time intervals?

The data preprocessing feature of transforming lat and long into grids can also be a method to leverage user position while maintaining some level of user privacy. This feature of the proposed work should be highlighted as a contribution.

The section Encoder-Decoder Prediction Model Based On Attention Mechanism is hard to follow. Please rewrite this section to increase its understandability. A suggestion is to list all the components in a bullet list at the beginning of the section to introduce: what are their inputs and outputs and how this component was chosen. Clarify how the output of the attention layer and the hidden layer vectors are combined into the decoder.

Regarding the performance evaluation, it only analyzed the overall PSAR algorithm. However, it utilizes the prediction of two algorithms: mobility and resource utilization. Other prediction techniques could replace the method used, and the PSAR algorithm would still function.

However, the efficiency of the proposed prediction algorithms must be evaluated. Quantifying its performance is paramount to assessing whether deep learning techniques (such as those utilized) are needed and what the trade-offs are when utilizing simpler prediction algorithms (you need good baselines).

Reviewer #5: Subject: Feedback on Article Evaluation

I hope this message finds you well. I have recently evaluated the article submitted to your esteemed magazine and would like to provide constructive feedback for your consideration.

The article delves into a significant debate crucial for the advancement of new-generation networks, a topic of considerable interest in current research. However, upon review, it is evident that the manuscript suffers from several critical issues that warrant attention.

1. **Abstract**:

- The abstract lacks a quantitative description to complement the qualitative evaluation presented.

- Algorithms compared with the proposed method should be explicitly mentioned.

- Address grammatical issues, and repetitions, and ensure clarity for enhanced coherence.

2. **Introduction**:

- Rectify grammatical errors and typos throughout the text.

- Improve clarity and coherence by enhancing organization and phrasing.

- Correct punctuation errors and unnecessary capitalizations.

- Condense repeated points for smoother flow.

- Define acronyms to aid readers' comprehension.

- Enhance engagement through more captivating language.

3. **Related Works**:

- Deepen the discussion of related works to address existing weaknesses.

- Categorize information for better structure and consider incorporating a table for advantages and disadvantages of works.

- Correct language errors and ensure consistency in terminology.

4. **System Model**:

- While the problem is well-formulated, analyze how the method overcomes identified limitations.

- Include an analysis of the algorithm's time complexity for a comprehensive understanding.

5. **Analysis of Simulation Results**:

- Address the following questions regarding the considered environment, user requirements, and method limitations.

- Evaluate the speed of convergence of machine learning algorithms and error functions.

- Update the dataset to reflect recent network service requirements.

- Consider the dynamic nature of the environment for more effective offline programming methods.

I believe addressing these concerns will significantly enhance the quality and impact of the article.

Warm regards,

Mohsen khani

6. PLOS authors have the option to publish the peer review history of their article (what does this mean?). If published, this will include your full peer review and any attached files.

Reviewer #1: No

Reviewer #2: **Yes: **Yicen Liu

Reviewer #3: No

Reviewer #4: **Yes: **Ivan Zyrianoff

Reviewer #5: **Yes: **Mohsen Khani

---

## [Author Response · Author response to Decision Letter 0]

19 May 2024

Reply to Reviewer #1

Problem 1: Clarify the specific performance metrics used to evaluate the SFC-RS, such as throughput, packet loss, or jitter, to provide a more comprehensive assessment of its effectiveness.

Response: Thank you very much for this comment. The performance metrics of SFC-RS are specifically stated in the Analysis of Performance Evaluation section of this paper. This paper evaluates the effectiveness of SFC-RS in terms of average SFC migration delay, reconfiguration cost, and physical node resource utilization. For detailed information, please see the context that are marked in red in section Performance Evaluation. Also, based on your suggestion, an analysis of throughput has been added to this article. For detailed information, please refer to Fig. 8 in the Performance Evaluation section and its analysis.

Problem 2: Provide a detailed overview of the existing algorithms or methodologies used for SFC reconfiguration in IoT-MEC networks to better contextualize the proposed PSAR algorithm.

Response: Thank you very much for this comment. I have revised the Introduction and Related work sections to evaluate existing SFC reconstruction algorithms. For detailed information, please see the red text in paragraph 3 of the Introduction and in paragraph 2, 5 of the related work.

Problem 3: Include a discussion on potential limitations or challenges associated with the proposed Encoder-Decoder mobile user trajectory prediction model and the VNF instance resource demand prediction model to address any uncertainties regarding their accuracy and reliability.

Response: Thank you very much for this comment. I have added a discussion of potential limitations or challenges associated with encoder-decoder mobile user trajectory prediction models and VNF instance resource demand prediction models to the conclusion section of this paper, please see the red text section of the Conclusion for detailed information.

Problem 3: Consider integrating additional machine learning techniques or algorithms to further enhance the accuracy of user trajectory prediction and node load estimation, potentially improving the overall performance of the PSAR algorithm.

Response: Thank you very much for this comment. We understand and agree with your suggestion of integrating additional machine learning techniques or algorithms to improve the accuracy of user trajectory prediction and node load estimation, which may to further improve the overall performance of the PSAR algorithm. However, considering the limited computational resources of existing servers, our PSAR algorithm has been realized to provide satisfactory prediction accuracy while maintaining low latency, which is crucial for real-world application scenarios. At the same time, we also hope to maintain the simplicity and interpretable of the model, in order to better understand the prediction mechanism and lay the foundation for future work. That is to say, we fully agree with exploring and integrating more advanced techniques such as integrated learning and meta-learning methods as part of future research. Thank you again for your valuable comments, and we look forward to the opportunity to continue to improve our work and contribute more to this research area.

Problem 4: Discuss the scalability of the proposed SFC reconfiguration strategy in terms of handling a large number of mobile IoT devices and VNF instances, particularly in scenarios with dynamically changing network conditions. Explore the potential impact of network topology changes, such as node failures or additions, on the effectiveness of the PSAR algorithm and investigate strategies for adaptability and resilience in such scenarios.

Response: Thank you very much for this comment. We have conducted in-depth consideration and analysis of the scalability of the SFC reconfiguration strategy in dealing with a large number of mobile IoT devices and VNF instances, as you mentioned. In designing the strategy, we considered the scalability required to cope with large-scale mobile IoT devices and VNF instances. The strategy is therefore modular in design and can effectively adapt to dynamically changing network environments. We also recognize that changes in network topology, especially the failure or addition of nodes, may challenge the effectiveness of the algorithm. Therefore, we have adapted our network topology settings to simulate network topology changes due to network node failures. For detailed dynamic topology setting parameters, please refer to the red text in paragraph 2 of the Performance Evaluation.

Problem 5: Provide insights into the computational complexity of the proposed ILP model and evaluate its scalability with increasing network size and traffic volume to ensure feasibility for real-world deployment.

Response: Thank you very much for this comment. We recognize that the computational efficiency of a model and its ability to handle large-scale problems are crucial in practical applications. Since the ILP problem in this paper is NP-hard problem, their solution time may increase exponentially with the problem size, Therefore, for the proposed ILP problem, we use experiments in the simulation section to evaluate the running time of the proposed algorithm in solving this ILP problem. For detailed information, please refer to Fig. 11 and its explanation.

Problem 6: Consider conducting sensitivity analyses or scenario-based simulations to assess the robustness of the PSAR algorithm under various network conditions and user mobility patterns.

Response: Thank you very much for your comment. In response to your suggestion, we have modified the simulation settings section to make the simulation scenarios closer to the real scenarios. Moreover, we use a series of randomly set server characteristics and network characteristics to simulate the changes of network conditions caused by user mobility, making our simulation experiments closer to reality. For detailed information, please refer to paragraph 2 of the Performance Evaluation section in the paper.

Problem 7: Address potential privacy concerns associated with the collection and processing of user trajectory data and VNF instance resource demand information, and propose measures to safeguard user privacy while maintaining prediction accuracy.

Response: Thank you very much for this comment. We adopt a data preprocessing method that converts latitude and length of user’s location to a grid for obtaining the user’s location while maintaining a certain degree of user privacy. For detailed information, please refer to paragraph 3 and 4 of the Data Pretreatment in the paper.

Problem 8: Discuss potential extensions or future research directions for enhancing the SFC reconfiguration strategy, such as incorporating edge computing capabilities or integrating dynamic service chaining mechanisms to adaptively respond to evolving network requirements.

Response: Thank you very much for your comments. In our current research, we have established a basic reconfiguration framework while recognizing that there are many potential extensions and future research directions to explore. For example, we adopt some new approaches to make our algorithms more adaptive to changing network demands, and we add a description of future research directions in the conclusion section, for example, the introduction of meta-learning is considered to improve the convergence speed as well as robustness of the model in this paper. Please see the red text in the Conclusion section for more details. 

Problem 9: Please avoid citing sources that were published before to 2019. Cite current research that are really pertinent to your topic. The study also lacks sufficient citations. Another critical step is to compare the topic of the article to other relevant recent publications or works in order to widen the research's repercussions beyond the issue. Authors can use and depend on these essential works while addressing the topic of their paper and current issues.

A. Heidari, N. J. Navimipour, M. A. J. Jamali, and S. Akbarpour, "A hybrid approach for latency and battery lifetime optimization in IoT devices through offloading and CNN learning," Sustainable Computing: Informatics and Systems, vol. 39, p. 100899, 2023/09/01/ 2023, doi: https://doi.org/10.1016/j.suscom.2023.100899.

Amiri, Z., Heidari, A., Navimipour, N.J. et al. Adventures in data analysis: a systematic review of Deep Learning techniques for pattern recognition in cyber-physical-social systems. Multimed Tools Appl (2023). https://doi.org/10.1007/s11042-023-16382-x

A. Heidari, N. J. Navimipour, M. A. J. Jamali, and S. Akbarpour, "A green, secure, and deep intelligent method for dynamic IoT-edge-cloud offloading scenarios," Sustainable Computing: Informatics and Systems, vol. 38, p. 100859, 2023.

A. Heidari, M. A. J. Jamali, N. J. Navimipour and S. Akbarpour, "A QoS-Aware Technique for Computation Offloading in IoT-Edge Platforms Using a Convolutional Neural Network and Markov Decision Process," in IT Professional, vol. 25, no. 1, pp. 24-39, Jan.-Feb. 2023, doi: 10.1109/MITP.2022.3217886.

Response: Thank you for your valuable comments and guidance. We fully agree with the importance of updating the literature citations, comparing the latest studies and expanding our research horizons. In order to improve the quality and relevance of the paper, we have thoroughly reviewed and revised the cited references. Considering the closed relevance to this paper, we cite the follows papers.

A. Heidari, N. J. Navimipour, M. A. J. Jamali, and S. Akbarpour, "A hybrid approach for latency and battery lifetime optimization in IoT devices through offloading and CNN learning," Sustainable Computing: Informatics and Systems, vol. 39, p. 100899, 2023/09/01/ 2023, doi: https://doi.org/10.1016/j.suscom.2023.100899.

A. Heidari, N. J. Navimipour, M. A. J. Jamali, and S. Akbarpour, "A green, secure, and deep intelligent method for dynamic IoT-edge-cloud offloading scenarios," Sustainable Computing: Informatics and Systems, vol. 38, p. 100859, 2023.

A. Heidari, M. A. J. Jamali, N. J. Navimipour and S. Akbarpour, "A QoS-Aware Technique for Computation Offloading in IoT-Edge Platforms Using a Convolutional Neural Network and Markov Decision Process," in IT Professional, vol. 25, no. 1, pp. 24-39, Jan.-Feb. 2023, doi: 10.1109 /MITP.2022.3217886. 

Reviewer #2: This paper proposes a novel SFC adjustment method, and simulation results verify the effectiveness of the method. The reviewer has the following comments and suggestions that might help the authors to improve their work.

Problem 1: The title of the paper is resource prediction, without specifying what specific resources are, node computing resources, memory resources, or link bandwidth resources.

Response: Thank you very much for this comment. In order to maintain the conciseness and brevity of the title, we did not clearly state what resources are in the title. However, we have stated in the Introduction section that it is the prediction of CPU and memory resource requirements. For detailed information, please refer to the red text in paragraph 4 of the Abstract section.

Problem 2: Generally speaking, SFC adjustments are made based on deployment, and the deployment mechanism often constrains the adjustment mechanism. How to consider the cost of some adjustment mechanisms being higher than the cost of redeployment?

Response: Thank you very much for your comment, in our proposed algorithm, SFC reconfiguration is triggered only when the load of a node exceeds a designated threshold. We can adjust the cost required to trigger the threshold by adjusting the size of the threshold. And as part of our optimization objective is the cost considered, we can also focus more on the cost item by changing the weight coefficient of the cost. 

Problem 3: The paper mainly considers the triggering factors of load balancing, and is it compatible with indicators such as latency?

Response: Thank you very much for this comment. In this paper, the reconfiguration triggering factor of SFC not only includes the load of nodes, but also takes into account the delay factor. As described in Algorithm 1, the reconfiguration of SFC is only triggered when the node's load exceeds the threshold. However, once the reconfiguration is triggered, the user's next position must be predicted through the user trajectory prediction module to estimate whether the reconfiguration delay of SFC meets the requirements. If it does not meet the requirements, SFC reconfiguration will not be carried out. For detailed information, please see the context that are marked in red of Algorithm 1 SFC Reconfiguration Trigger Algorithm.

Problem 4: In the literature analysis of the paper, the comparison algorithm did not consider IoT user mobility. However, does the author's use of it as a comparison affect the fairness of the comparison?

Response: Thank you very much for this comment. The comparison algorithm DPSM [22] considered the user’s movement trajectory. We have revised the explanation of the DPSM. For detailed information, please see the context that are marked in red of Comparison Algorithms.

Problem 5: During the experiment, there was a lack of validation of the effectiveness of the predictive model.

Response: Thank you very much for your comments. We add the explanation of the prediction model. For more detailed information, please see the red text in paragraph 1 of the Simulation Settings. Additionally, in the simulation validation, we used comparisons with algorithms without predictions to verify the validity of the predictive model. For more details, please refer to Fig. 9 and the text marked in red in its analysis.

->It is suggested to refer to the related work, such as

Forecast-Assisted Service Function Chain Dynamic Deployment for SDN/NFV-Enabled Cloud Management Systems, IEEE SY-J,2023

SFC Embedding Meets Machine Learning: Deep Reinforcement Learning Approaches, IEEE COMML, 2021

Service Function Chain Embedding Meets Machine Learning: Deep Reinforcement Learning Approach, IEEE TNSM,2024.

Response: Thank you very much for your comments. We have cited the follows paper.

Forecast-Assisted Service Function Chain Dynamic Deployment for SDN/NFV-Enabled Cloud Management Systems, IEEE SY-J,2023

SFC Embedding Meets Machine Learning: Deep Reinforcement Learning Approaches, IEEE COMML, 2021

Reviewer #3: 

Problem 1: Some existing works have studied similar problem in edge computing by proposing similar methods, what are main differences between the proposed algorithm and existing works.

Response: Thank you for your careful review and valuable suggestions. Compared with existing studies, our algorithm not only considers user mobility but also integrates the prediction of resource demand and user location when dealing with SFC reconfiguration strategies in IoT-MEC networks. To the best of our knowledge, existing studies generally deal with these factors separately and rarely jointly consider them. For detailed information, please see the red text in the para. 3 of the Introduction section.

Problem 2: There are a number of grammar errors and typos in this manuscript. The authors have to carefully revise this manuscript before submission. For example,

“The simulation results show that, Compared with … ”

Response: Thank you for your comments. We have carefully reviewed the full text and revised the grammar errors and typos to ensure a fluent and logical presentation. We are committed to eliminating these issues completely in our revised manuscripts.

Problem 3: Check to use uniform expression formats in context, such as equation(x) and equation x.

Response: Thank you for your comments. We have carefully reviewed the full text and made the necessary changes to ensure that all mathematical equations and related expressions are uniformly formatted according to academic standards.

Problem 4: Please improve the experimental parameter settings and algorithm analysis to demonstrate the superiority of the algorithm in practical use.

Response: Thank you for your comments. In order to prove the superiority of the algorithm in practical applications, we have adjusted and optimized the experimental parameter settings and rerun the partial simulation experiments. For detailed information, pl

---

## [Decision Letter · Decision Letter 1]

24 Jun 2024

SFC Active Reconfiguration based on User Mobility and Resource Demand Prediction in Dynamic IoT-MEC Networks

PONE-D-24-02486R1

Dear Dr. Liu,

We’re pleased to inform you that your manuscript has been judged scientifically suitable for publication and will be formally accepted for publication once it meets all outstanding technical requirements.

Kind regards,

Muhammad Anwar, Ph.D.

Academic Editor

PLOS ONE

Additional Editor Comments (optional):

Comments from PLOS Editorial Office: We note that one or more reviewers has recommended that you cite specific previously published works in an earlier round of revision. As always, we recommend that you please review and evaluate the requested works to determine whether they are relevant and should be cited. It is not a requirement to cite these works and you may remove them before the manuscript proceeds to publication. We appreciate your attention to this request.

Reviewers' comments:

Reviewer's Responses to Questions

**Comments to the Author**

1. If the authors have adequately addressed your comments raised in a previous round of review and you feel that this manuscript is now acceptable for publication, you may indicate that here to bypass the “Comments to the Author” section, enter your conflict of interest statement in the “Confidential to Editor” section, and submit your "Accept" recommendation.

Reviewer #1: All comments have been addressed

Reviewer #4: All comments have been addressed

Reviewer #5: All comments have been addressed

2. Is the manuscript technically sound, and do the data support the conclusions?

Reviewer #1: Yes

Reviewer #4: Yes

Reviewer #5: Partly

3. Has the statistical analysis been performed appropriately and rigorously? 

Reviewer #1: Yes

Reviewer #4: Yes

Reviewer #5: I Don't Know

4. Have the authors made all data underlying the findings in their manuscript fully available?

Reviewer #1: Yes

Reviewer #4: Yes

Reviewer #5: Yes

5. Is the manuscript presented in an intelligible fashion and written in standard English?

Reviewer #1: Yes

Reviewer #4: Yes

Reviewer #5: Yes

6. Review Comments to the Author

Reviewer #1: Authors properly revised the paper. Thank you for their efforts. So, it can be accepted now without any change.

Reviewer #4: The authors addressed all my comments. Thus, I recommend the publication of this work.

A minor comment is that in the abstract it is listed other methods with the authors compared their work (TSRFCM, DDQ, OSA, and DPSM), but those initials were not introduced and are not well known. I recommend the authors to rewrite to "Simulation results

show that PSAR provides 51.28$\\%$, 28.60$\\%$, 21.75$\\%$, and 16.80$\\%$ performance improvement over the existing state-of-art algorithms in terms of end-to-end delay reduction" and then clarify what are those algorithms in the introduction.

Reviewer #5: Please refer to recent articles published in the relevant field in related works;

Some examples of work are:

Resource allocation in 5G cloud-RAN using deep reinforcement learning algorithms: A review

An enhanced deep reinforcement learning-based slice acceptance control system (EDRL-SACS) for cloud-radio access network

Three-layer data center-based intelligent slice admission control algorithm for C-RAN using approximate reinforcement learning

7. PLOS authors have the option to publish the peer review history of their article (what does this mean?). If published, this will include your full peer review and any attached files.

Reviewer #1: **Yes: **Arash Heidari

Reviewer #4: **Yes: **Ivan Zyrianoff

Reviewer #5: No

---

## [Editor Report · Acceptance letter]

28 Jun 2024

PONE-D-24-02486R1 

PLOS ONE

Dear Dr. Liu, 

I'm pleased to inform you that your manuscript has been deemed suitable for publication in PLOS ONE. Congratulations! Your manuscript is now being handed over to our production team.

Kind regards, 

on behalf of

Dr. Muhammad Anwar 

Academic Editor

PLOS ONE